# E2F1 acetylation directs p300/CBP-mediated histone acetylation at DNA double-strand breaks to facilitate repair

Swarnalatha Manickavinayaham [1], Renier Vélez-Cruz[1,3], Anup K. Biswas[1,4], Ella Bedford[1], Brianna J. Klein[2], Tatiana G. Kutateladze [2], Bin Liu[1], Mark T. Bedford [1] & David G. Johnson [1]*

E2F1 and retinoblastoma (RB) tumor-suppressor protein not only regulate the periodic expression of genes important for cell proliferation, but also localize to DNA double-strand breaks (DSBs) to promote repair. E2F1 is acetylated in response to DNA damage but the role this plays in DNA repair is unknown. Here we demonstrate that E2F1 acetylation creates a binding motif for the bromodomains of the p300/KAT3B and CBP/KAT3A acetyltransferases and that this interaction is required for the recruitment of p300 and CBP to DSBs and the induction of histone acetylation at sites of damage. A knock-in mutation that blocks E2F1 acetylation abolishes the recruitment of p300 and CBP to DSBs and also the accumulation of other chromatin modifying activities and repair factors, including Tip60, BRG1 and NBS1, and renders mice hypersensitive to ionizing radiation (IR). These findings reveal an important role for E2F1 acetylation in orchestrating the remodeling of chromatin structure at DSBs to facilitate repair.

[1] Department of Epigenetics and Molecular Carcinogenesis, The University of Texas MD Anderson Cancer Center, Science Park, Smithville, TX 78957, USA. [2] Department of Pharmacology, University of Colorado School of Medicine, Aurora, CO 80045, USA. [3] Present address: Department of Biochemistry, Midwestern University, Chicago College of Osteopathic Medicine, Downers Grove, IL 60515, USA. [4] Present address: Herbert Irving Comprehensive Cancer Center, Columbia University, New York, NY 10032, USA. *email: djohnson@mdanderson.org

The E2F1 transcription factor regulates the expression of genes involved in cell proliferation, apoptosis, and other cellular processes[1,2]. An important regulator of E2F1 is the retinoblastoma (RB) tumor-suppressor protein, which can bind and convert E2F1 from an activator to a repressor of transcription. E2F1 activity is deregulated in most cancers by disruptions in the RB pathway but the role of E2F1 in cancer is complex as it can either promote or inhibit tumor development depending on the context[2]. In addition to regulating transcription, a number of studies have revealed transcription-independent functions for E2F1 and RB in regulating DNA repair[3–7]. E2F1 localizes to sites of DNA damage dependent on its phosphorylation at serine 31 (serine 29 in mice) by the ATM or ATR kinases[8,9]. The topoisomerase II binding protein (TopBP1), which has multiple functions during the DNA damage response, specifically binds this phosphorylated form of E2F1 resulting in recruitment of E2F1 to sites of DNA damage[6,8]. RB associates with E2F1 at DNA double-strand breaks (DSBs) and helps to stabilize the interaction between phosphorylated E2F1 and TopBP1[7]. Thus there is a mutual dependence between E2F1 and RB for their recruitment to DSBs. Consistent with this finding, the absence of RB causes the same defects in DNA end-resection and homologous recombination (HR) as is observed in the absence of E2F1[7]. A knock-in mutation that prevents E2F1 phosphorylation (S29A) and recruitment of E2F1 and RB to sites of DNA breaks also impairs DNA repair and renders mice hypersensitive to ionizing radiation (IR)[7].

In addition to phosphorylation, E2F1 is also acetylated in response to DNA damage on three lysine residues (K117, K120, and K125 in human E2F1) near the DNA-binding domain. Different reports have implicated PCAF/KAT2B[10,11], CBP/KAT3A and p300/KAT3B[12,13] and Tip60/KAT5[14] in mediating E2F1 acetylation. Multiple deacetylases, including histone deacetylase 1 (HDAC1)[13] and SirT1[15], can also regulate E2F1 acetylation levels. E2F1 acetylation can occur in the absence of E2F1 phosphorylation and independently promotes E2F1 stabilization in response to DNA damage and may also target E2F1 to specific pro-apoptotic genes, such as p73[16]. Notably, how acetylation regulates the DNA repair function of E2F1 has not been addressed.

Bromodomains specifically recognize and bind acetylated lysine motifs and are found in a number of proteins associated with chromatin and involved in transcriptional regulation and/or DNA repair[17]. Small molecule inhibitors of bromodomains, such as those targeting the BET family of proteins, are emerging as promising drugs for cancers and other diseases[17,18]. It is therefore important to identify the acetylated protein targets that are read and regulated by bromodomain-containing proteins.

Here we demonstrate that the bromodomains of the related p300 and CBP acetyltransferases specifically bind to the acetylated motif of E2F1. This interaction with acetylated E2F1 is critical for p300/CBP recruitment to DSBs and induction of histone acetylation at the sites of damage. Mutating the three sites of E2F1 acetylation also impairs the recruitment of other chromatin-modifying enzymes and repair factors to DSBs, leads to defective DNA repair, and renders mice hypersensitive to IR. These findings define a mechanism by which p300 and CBP are recruited to DSBs dependent on E2F1 and RB and reveal crosstalk between E2F1 acetylation and histone acetylation as an important component of the DNA damage response.

## Results

### The bromodomains of p300 and CBP bind to acetylated E2F1.
In response to DNA damage, human E2F1 is acetylated on lysine residues K117, K120, and K125[10,12,14,16]. To identify proteins that interact with this acetylated form of E2F1, we screened a protein domain microarray with peptides that were either unacetylated or acetylated on lysine residues corresponding to K117, K120, and K125 of human E2F1 (Fig. 1a and Supplementary Fig. 1a). The bromodomain of p300 was identified as binding to the E2F1 peptide when acetylated but not unacetylated (Fig. 1a). Peptide pull-down assays using glutathione S-transferase (GST) fusion proteins confirmed a specific interaction between the acetylated E2F1 peptide and the bromodomain of p300 (Fig. 1b) as well as the bromodomain of the closely related acetyltransferase, CBP (Fig. 1c). The binding was further corroborated by $^1$H,$^{15}$N heteronuclear single quantum coherence (HSQC) nuclear magnetic resonance (NMR) titration experiments (Fig. 1d). The E2F1ac peptide induced chemical shift perturbations in either the isolated bromodomain of p300 or the p300 construct containing bromodomain, the RING finger, and PHD finger (BRP), whereas unacetylated E2F1 peptide failed to do so (Supplementary Fig. 1b). Further analysis using mono-acetylated and di-acetylated E2F1 peptides indicates that acetylation at K125 is sufficient for binding the bromodomains of p300 and CBP (Fig. 1b, c and Supplementary Fig. 1c). However, GST-CBP, but not GST-p300, was also efficiently pulled down using a K117 mono-acetylated peptide (Fig. 1b, c).

E2F1 is recruited to DSB sites via a phospho-specific interaction with the BRCT domain-containing protein TopBP1[8]. We previously used a recombinant GST-TopBP1 fusion protein to specifically pull-down the phosphorylated form of E2F1 and its associated proteins[7]. In addition to pulling down E2F1 and RB, the GST-TopBP1 fusion construct also pulled down endogenous p300 and CBP from extracts made from wild-type mouse embryonic fibroblasts (MEFs) treated with IR but not from untreated MEFs (Fig. 2a). A homozygous E2f1 S29A knock-in mutation that prevents E2F1 phosphorylation and its interaction with TopBP1 also prevents association of p300 and CBP with the GST-TopBP1 fusion construct (Fig. 2a). GST-TopBP1 also pulled down endogenous p300 and CBP, along with E2F1 and RB, from extracts made from human U2OS cells treated with IR but not from untreated cell extract (Fig. 2b). As we previously observed[7], knocking down RB reduced the interaction between E2F1 and TopBP1 and also prevented the IR-inducible association of p300 and CBP with GST-TopBP1. This suggests that RB helps to stabilize the interaction between p300/CBP and the phosphorylated form of E2F1 that is recognized by TopBP1.

**E2F1 recruits p300 and CBP to DNA DSBs.** Previous studies demonstrated that p300 and CBP are recruited to DNA breaks and participate in local histone acetylation and remodeling of chromatin structure to facilitate repair[19–21]. However, the mechanism by which p300 and CBP are recruited to DSBs is not fully understood. We previously used an inducible I-PpoI endonuclease system[22] combined with chromatin immunoprecipitation (ChIP) to demonstrate that E2F1 and RB are enriched at DNA sequences flanking DSBs dependent on E2F1 phosphorylation by ATM[7]. Using this assay, we confirmed that E2F1 and RB are recruited to an I-PpoI-induced DSB in mouse chromosome 5 (mChrom5) in primary wild-type MEFs but not in $E2f1^{S29A/S29A}$ MEFs (Fig. 3a). In contrast, γH2AX is enriched at the induced DSB in both wild-type and $E2f1^{S29A/S29A}$ MEFs. Consistent with our finding that p300 and CBP associate with E2F1 in response to DNA damage, p300 and CBP were also recruited to the induced DSB in wild-type MEFs but not in MEFs harboring the E2f1 S29A mutation (Fig. 3a). Moreover, H3K18ac and H3K56ac, two histone acetylation marks generated by p300/CBP[23–27], were enriched at the DSB in wild-type but not in S29A knock-in MEFs. This defect in p300 and CBP recruitment in

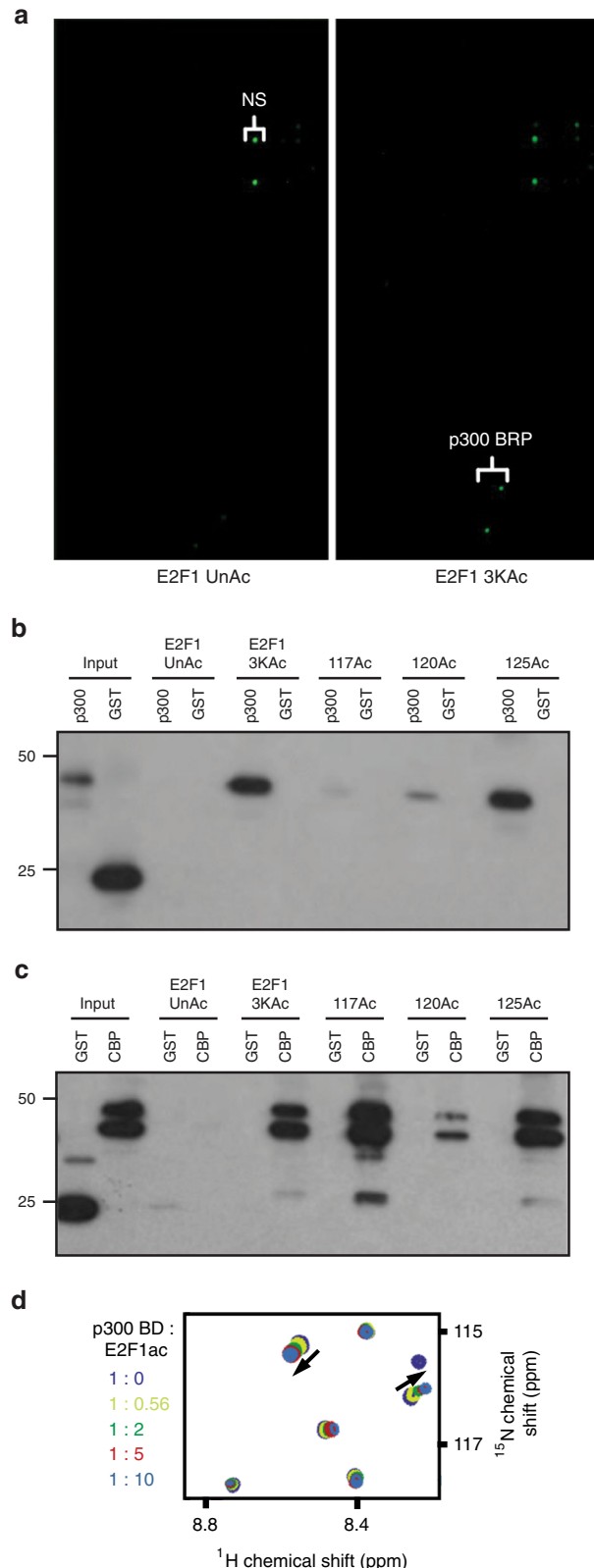

**Fig. 1** E2F1 acetylation creates a binding motif for the bromodomains of p300 and CBP. **a** A tri-acetylated E2F1 peptide (3KAc) but not unacetylated peptide (UnAc) binds to recombinant p300 containing the bromo, ring finger, and PHD domains (BRP) in a protein domain microarray. NS indicates nonspecific interaction. **b**, **c** E2F1 peptides unacetylated, tri-acetylated, or mono-acetylated at K117, K120, or K125 were incubated with purified GST-fusion proteins containing the bromodomain domain of p300 (**b**) or CBP (**c**). Following pull-down, western blot for GST was performed to examine binding between p300 or CBP and the E2F1 peptides. **d** A $^1$H, $^{15}$N heteronuclear single quantum coherence NMR titration experiment was performed using the p300 bromodomain and an acetylated E2F1 peptide. Source data of **b** and **c** are provided as Supplementary Data 5

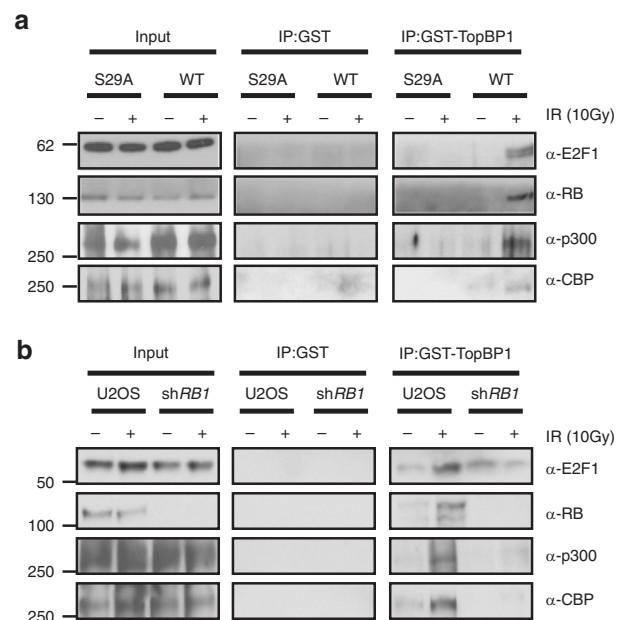

**Fig. 2** Phosphorylated E2F1 interacts with p300 and CBP in response to DNA damage. **a** GST-TopBP1 (BRCT1-6) or GST control protein were incubated with whole-cell extract from wild-type (WT) or $E2f1^{S29A/S29A}$ (S29A) MEFs that were either untreated (−) or treated (+) with IR (10 Gy) and associated proteins were identified 2 h post-IR by western blot analysis. **b** A similar GST-TopBP1 pull-down assay was performed using extracts from parental U2OS cells or cells knocked down for RB, either untreated (−) or treated (+) with IR (10 Gy). Source data of **a** and **b** are provided as Supplementary Data 5

$E2f1^{S29A/S29A}$ MEFs is not due to differences in E2F1, RB, p300, or CBP protein levels (Supplementary Fig. 2a). No enrichment of E2F1, RB, p300, CBP, or H3 acetylation marks was observed at the *Gapdh* locus, which lacks an I-PpoI cut site (Supplementary Fig. 2b).

The I-PpoI cut site in mChrom5 analyzed above is located in an euchromatic region, 5' to *RIKEN cDNA 4930519G04* gene. We

previously demonstrated that E2F1 and RB are also recruited to an I-PpoI-induced DNA break in a heterochromatic region of mouse chromosome 10 (mChrom10)[7]. Both p300 and CBP were also enriched at this chromosome 10 cut site following I-PpoI induction in wild-type MEFs, as were several H3 acetylation marks mediated by these enzymes (Supplementary Fig. 2c). However, p300 and CBP recruitment and induction of H3 acetylation at this DSB were abolished in $E2f1^{S29A/S29A}$ knock-in MEFs even though γH2AX was enriched as in wild-type MEFs. Using a similar I-PpoI assay system in human U2OS cells, we find that E2F1, RB, p300, and CBP are also recruited to a DSB induced within the *DAB1* gene on chromosome 1[28] and this was associated with an induction of H3K56 acetylation (Fig. 3b). Depletion of E2F1 or RB in U2OS cells did not affect p300 or CBP protein levels (Supplementary Fig. 3a) or phospho-ATM enrichment at the DNA break site, but it did prevent enrichment of p300, CBP, and H3K56ac (Fig. 3b). Similar results were observed at the *rDNA* locus in U2OS cells but not at the control *GAPDH* locus (Supplementary Fig. 3b, c). Taken together, these findings

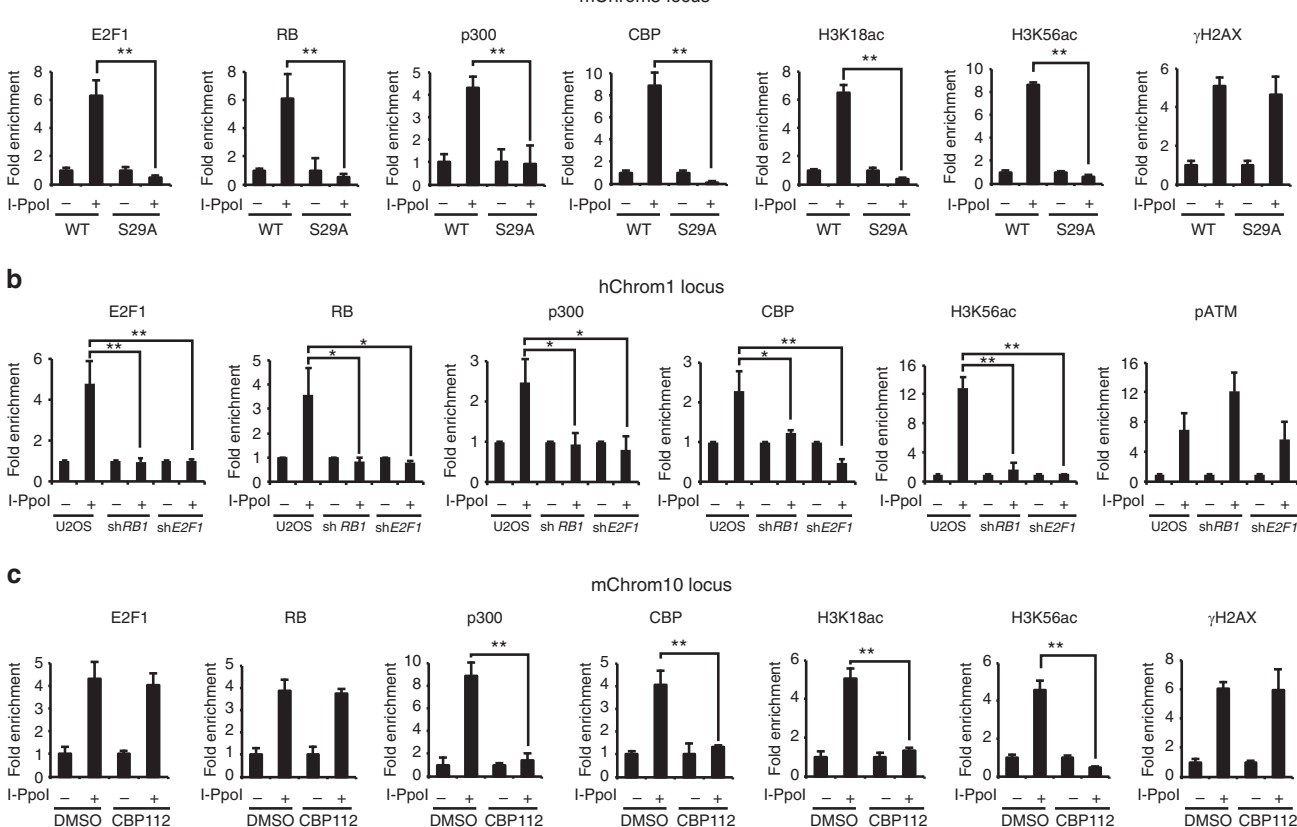

**Fig. 3** Recruitment of p300 and CBP to DSBs is dependent on E2F1 and RB. **a** Primary wild-type (WT) or *E2f1*[S29A/S29A] (S29A) MEFs were uninfected (−) or infected (+) with a retrovirus expressing HA-ER*-I-PpoI and induced with 2 μM 4-hydroxy tamoxifen (4-OHT) for 12 h. ChIP was performed for E2F1, RB, p300, CBP, H3K18ac, H3K56ac, and γH2AX as indicated. Occupancy was determined by qPCR for a region on mouse chromosome 5 (mChrom5), 166 bp 3′ to an I-PpoI cut site. Fold enrichment represents the ratio between percentage of input of infected cells and uninfected cells. **b** A similar I-PpoI ChIP assay was performed and analyzed as above using parental U2OS cells and U2OS cells expressing shRNA to *RB1* and *E2F1*. Occupancy of E2F1, RB, p300, CBP, H3K56ac, and phospho-ATM was determined for a region on human chromosome 1 (hChrom1), 280 bp 5′ to I-PpoI cut site. **c** Primary wild-type MEFs either untreated (DMSO vehicle) or treated with the bromodomain inhibitor CBP112 (1 μM for 24 h) were subjected to the I-PpoI ChIP assay as above and occupancy was determined by qPCR at the mouse chromosome 10 (mChrom10) locus. Graphs represent average ± SD of three independent experiments (*n* = 3). *P* values were calculated by unpaired Student's *t* test. **P ≤ 0.01 is highly significant and *P ≤ 0.05 is significant. Raw data of **a–c** are provided as Source Data File

reveal that p300 and CBP are recruited to DSBs in different chromatin environments dependent on E2F1 and RB and that this is critical for inducing several H3 acetylation marks at the sites of damage.

Incubation of wild-type MEFs with CBP112, a small molecule inhibitor specific for the bromodomains of CBP and p300[29], did not affect γH2AX induction or recruitment of E2F1 and RB to an I-PpoI-induced DSB but it did inhibit recruitment of p300 and CBP and induction of H3K18ac and H3K56ac at the site of damage (Fig. 3c). Western blot analysis demonstrated that CBP112 treatment had little effect on the total protein levels of p300 and CBP or occupancy of p300 and CBP at the negative control *Gapdh* locus (Supplementary Fig. 3d, e). This indicates that the bromodomain inhibitor specifically blocks enrichment of p300 and CBP at DSBs and highlights the importance of the bromodomains for their E2F1-dependent recruitment to DNA breaks.

**Generation of *E2f1* acetylation-deficient mouse model**. We generated a targeted mutant mouse line in which the three sites of E2F1 acetylation were mutated from lysine to arginine and named this allele *E2f1*[3KR] (Supplementary Fig. 4a, b). Similar to

*E2f1*[−/−] and *E2f1*[S29A/S29A] knock-in mouse models, homozygous *E2f1*[3KR/3KR] knock-in mice are viable and display no obvious phenotype under unstressed conditions. RNA-sequencing (RNA-seq) analysis was performed on wild-type and *E2f1*[3KR/3KR] MEFs, before and after treatment with the radiomimetic drug neo-carzinostatin (NCS), to examine the impact of the 3KR mutation on global gene expression patterns. Expression levels of *E2f1* were similar between wild-type and *E2f1*[3KR/3KR] MEFs, consistent with similar E2F1 protein levels in wild-type and 3KR knock-in cells before and after DNA damage (Fig. 4a, b). As expected, the 3KR mutation prevented E2F1 acetylation in response to DNA damage (Fig. 4c).

A previous study suggested that E2F1 acetylation is involved in regulating the *p73* gene promoter in response to DNA damage[16]. In primary MEFs, however, expression of *p73* was barely detectable with very few sequence tags in each sample, independent of treatment or genotype (Fig. 4a). In sharp contrast, the expression of *Apaf1* and *Caspase3*, two other pro-apoptotic genes regulated by E2F[30,31], are readily detectable in primary MEFs and their expression increases in response to DNA damage (Fig. 4a). The basal expression level of *Apaf1*, but not of *Caspase3*, was lower in *E2f1*[3KR/3KR] MEFs, but induction in response to DNA damage was similar between genotypes for both genes, and

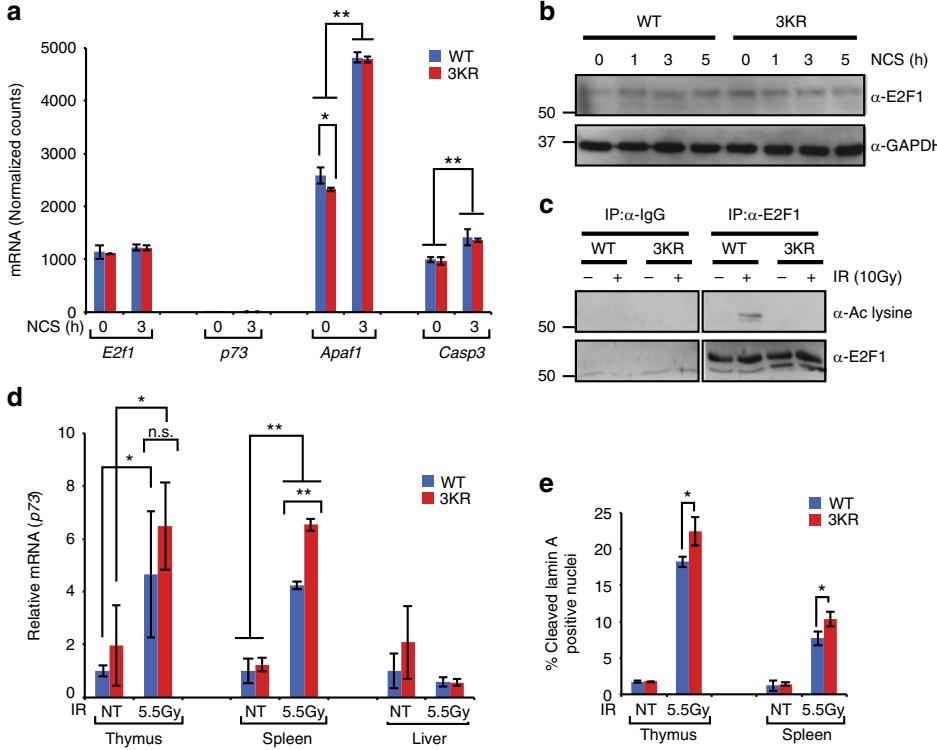

**Fig. 4** Characterization of the E2F1 acetylation 3KR knock-in mouse model. **a** RNA-seq was performed on wild-type (WT) and *E2f1³ᴷᴿ/³ᴷᴿ* (3KR) MEFs untreated or treated with NCS (250 ng/ml for 3 h) and the normalized mRNA counts for *E2f1*, *p73*, *Apaf1*, and *Caspase3* (*Casp3*) are presented. **b** Western blot analysis was performed to examine E2F1 protein levels in primary wild-type (WT) and *E2f1³ᴷᴿ/³ᴷᴿ* (3KR) MEFs before and after DNA damage with NCS (250 ng/ml) for the indicated times. **c** Wild-type and *E2f1³ᴷᴿ/³ᴷᴿ* MEFs were untreated or exposed to 10 Gy IR and cell extracts made 2 h later. Co-immunoprecipitation was performed using control IgG or antibody to E2F1 (C-20, rabbit polyclonal) followed by western blot analysis for acetylated lysine (top) or E2F1 (bottom) using a different antibody (KH95, mouse monoclonal). **d** Wild-type (WT) and *E2f1³ᴷᴿ/³ᴷᴿ* (3KR) mice were exposed to 5.5 Gy of IR and tissues collected 2 h later. Total RNA was isolated and cDNA was prepared by RT-PCR using random hexamer primer. Expression of *p73* in the thymus, spleen, and liver was determined by qPCR. **e** Wild-type (WT) and *E2f1³ᴷᴿ/³ᴷᴿ* (3KR) mice were exposed to 5.5 Gy of IR and thymus and spleen tissues were collected 2 h later. Tissue sections were stained for the cleaved form of Lamin A. The average of percentage of positive nuclei is presented. All graphs represent average ± SD of three independent experiments (*n* = 3) of cells or mice. *P* values were calculated by unpaired Student's *t* test. **P ≤ 0.01 is highly significant and *P ≤ 0.05 is significant. Source data of **b** and **c** are provided as Supplementary Data 5. Raw data of **a**, **d**, and **e** are in Source Data File

this was confirmed by reverse transcriptase real-time quantitative PCR (RT-qPCR) analysis of independent samples (Supplementary Fig. 4c). To determine whether *p73* expression and induction in response to DNA damage is tissue restricted, *E2f1³ᴷᴿ/³ᴷᴿ* and wild-type control mice were untreated or exposed to 5.5 Gy of IR and RNA was isolated from various tissues 2 h post-IR. The expression of *p73* was significantly induced in the thymus and spleen, but not in the liver of irradiated mice. However, contrary to expectations, the E2F1 3KR mutation actually enhanced the induction of *p73* expression in response to IR, although this was significant only in the spleen (Fig. 4d). Consistent with this finding, the apoptotic response to IR in the thymus and spleen was higher in *E2f1³ᴷᴿ/³ᴷᴿ* mice compared to wild-type mice (Fig. 4e).

Gene Set Enrichment Analysis (GSEA) was used to analyze the RNA-seq data from primary MEFs to determine which functional pathways might be altered by the E2F1 3KR mutation. Fifty-five out of 2499 gene sets were significantly enriched (upregulated) in wild-type cells compared to *E2f1³ᴷᴿ/³ᴷᴿ* mutant cells before DNA damage (Supplementary Data 1) and 96 gene sets were enriched in wild-type cells compared to *E2f1³ᴷᴿ/³ᴷᴿ* cells after DNA damage (Supplementary Data 2). The majority of gene sets whose expression was significantly lower in *E2f1³ᴷᴿ/³ᴷᴿ* mutant cells compared to wild-type cells, either before or after DNA damage, were related to neurodevelopment and/or differentiation. Almost 200 gene sets were found to be significantly enriched in *E2f1³ᴷᴿ/*

*³ᴷᴿ* cells compared to that in wild-type cells in the absence or presence of DNA damage, respectively. Gene sets upregulated in *E2f1³ᴷᴿ/³ᴷᴿ* cells were primarily involved in innate immune function (Supplementary Data 3 and 4). Importantly, no gene sets related to DNA repair, DNA damage response, cell cycle, or apoptosis were significantly different between wild-type and *E2f1³ᴷᴿ/³ᴷᴿ* knock-in MEFs, before or after DNA damage. These findings indicate that, while the E2F1 3KR mutation has a significant impact on global gene expression patterns, even in the absence of DNA damage, genes involved in DNA damage repair and cell cycle checkpoint responses are not significantly affected.

**E2F1 3KR mutation impairs histone acetylation at DSBs.** To determine how mutating the sites of E2F1 acetylation impacts its recruitment to DSBs, as well as the recruitment of other factors, we performed the I-PpoI ChIP assay in primary MEFs derived from homozygous *E2f1³ᴷᴿ/³ᴷᴿ* knock-in and wild-type control mice. The E2F1 3KR mutation did not significantly impact induction of γH2AX or enrichment of E2F1 and RB at a DSB located on mChrom10 but recruitment of p300 and CBP and induction of H3K18ac and H3K56ac were abolished (Fig. 5a). This difference in recruitment was not due to differences in p300 and CBP protein levels between genotypes before or after DNA damage induction (Fig. 5b). Similar results were obtained at another I-PpoI cut site on mChrom5 (Supplementary Fig. 5a). No

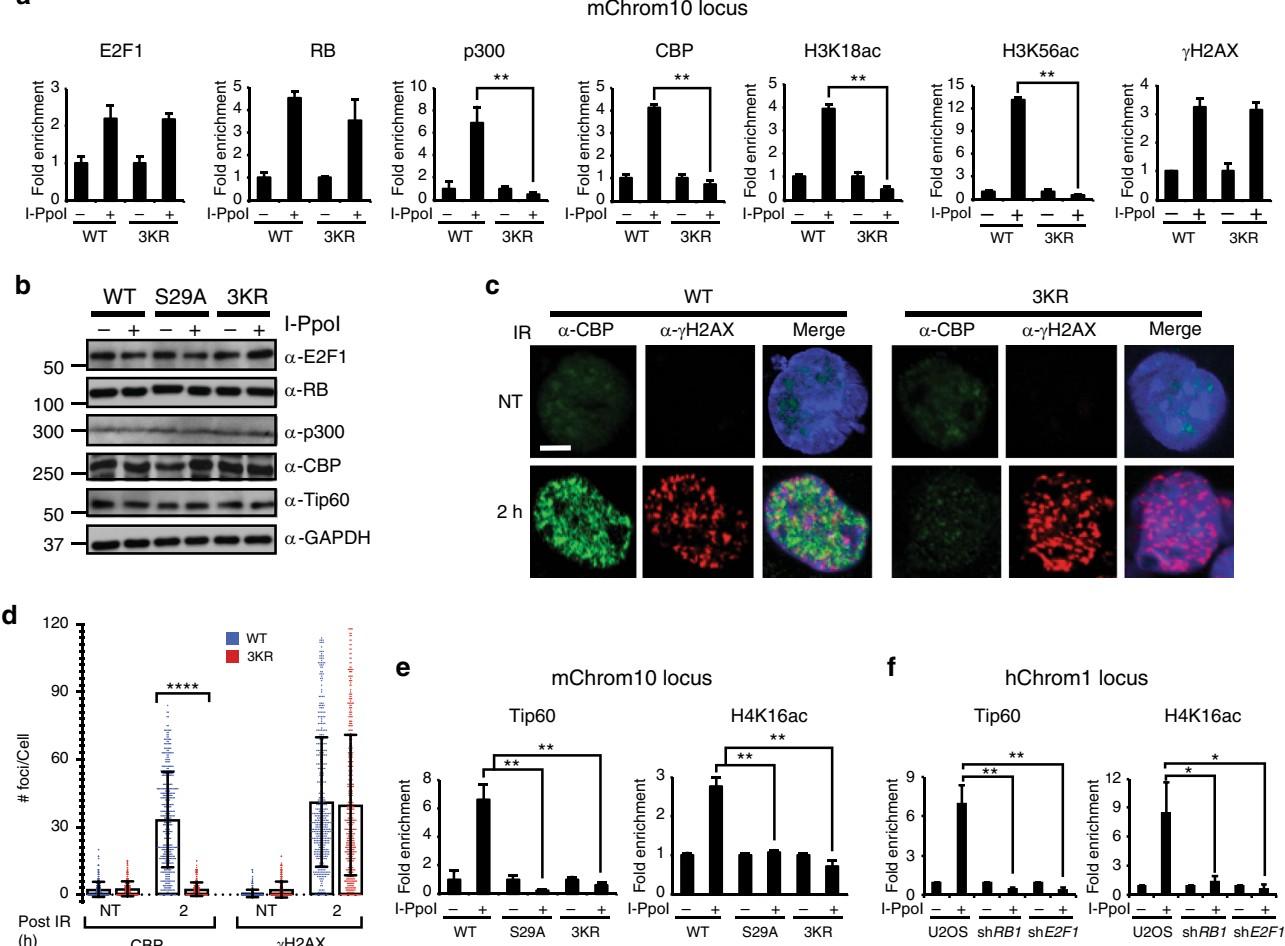

**Fig. 5** The E2F1 3KR mutation impairs multiple histone acetylation events at DSBs. **a** Primary wild-type (WT) and *E2f1³ᴷᴿ/³ᴷᴿ* (3KR) MEFs were subjected to the I-PpoI ChIP assay and qPCR was performed to determine the occupancy of the indicated proteins and H3 acetylation marks at mChrom10 locus. **b** Wild-type (WT), *E2f1ˢ²⁹ᴬ/ˢ²⁹ᴬ* (S29A), and *E2f1³ᴷᴿ/³ᴷᴿ* (3KR) MEFs were uninfected or infected with a retrovirus expressing inducible I-PpoI, treated with 4-OHT, and western blot analysis was performed for E2F1, RB, p300, CBP, Tip60, and GAPDH. **c** Wild-type and *E2f1³ᴷᴿ/³ᴷᴿ* MEFs were mock treated (NT) or exposed to 5 Gy of IR 2 h prior to in situ extraction and fixation. Representative images show formation of IR-induced foci for CBP (green) and γH2AX (red) and merged with DAPI staining (blue) of nuclei. Bar, 10 μm. **d** Quantification of CBP and γH2AX foci. Each treatment group was in triplicate, and in total at least 450 cells (*n* > 450) were counted per treatment group. Graphs represent average ± SD of foci per cell. Overlaid scattered dot plot shows the distribution of foci. *P* values were calculated by unpaired Mann–Whitney *U* test. ****$P < 0.0001$ is highly significant. **e** Primary wild-type (WT), *E2f1ˢ²⁹ᴬ/ˢ²⁹ᴬ* (S29A), and *E2f1³ᴷᴿ/³ᴷᴿ* (3KR) MEFs were subjected to the I-PpoI ChIP assay and occupancy of Tip60 and H4K16ac at mChrom10 locus was determined. **f** Parental U2OS cells and U2OS cells expressing shRNA to *RB1* and *E2F1* were subjected to the I-PpoI ChIP assay and qPCR was performed to determine occupancy of Tip60 and H4K16ac at hChrom1 locus. Except **d**, other graphs represent average ± SD of three independent experiments (*n* = 3). *P* values were calculated by unpaired Student's *t* test. **$P ≤ 0.01$ is highly significant and *$P ≤ 0.05$ is significant. Source data of **b** is provided as Supplementary Data 5. Raw data of **a** and **d**–**f** are in Source data file

enrichment of p300, CBP, or histone H3 acetylation marks was observed in either MEF culture after I-PpoI induction at the negative control *Gapdh* locus (Supplementary Fig. 5b). These findings are consistent with the bromodomain inhibitor data and the model in which phosphorylation recruits E2F1 and RB to DNA breaks while E2F1 acetylation is recognized by p300 and CBP leading to their recruitment and induction of H3 acteylation at the sites of DNA damage.

To further support this model, immunofluorescence (IF) for CBP and p300 was performed to observe IR-induced foci formation as an indicator of recruitment to DSBs. CBP and p300 both formed foci in response to IR that partially co-localized with γH2AX in wild-type MEFs (Fig. 5c, d and Supplementary Fig. 5c). However, the E2F1 3KR mutation abolished IR-induced foci formation of p300 and CBP while γH2AX induction was unaffected. These findings again indicate an important function

for E2F1 acetylation in the recruitment of p300/CBP to damaged DNA.

In addition to H3 acetylation, histone H4 acetylation is also known to be induced at DSBs by the Tip60 acetyltransferase and this is important for DNA repair and cell cycle checkpoint signaling[20,32–34]. As expected, ChIP analysis revealed Tip60 and H4K16ac enrichment at an I-PpoI-induced break on mChrom10 in wild-type MEFs (Fig. 5e). However, mutating the sites of E2F1 phosphorylation (S29A) or acetylation (3KR) prevented Tip60 recruitment and H4K16ac induction at the DSB site (Fig. 5e). Similar results were observed at another I-PpoI-induced cut site on mChrom5 but not at the negative control *Gapdh* locus (Supplementary Fig. 6a, b).

Depleting E2F1 or RB in U2OS cells also impaired Tip60 recruitment and induction of H4K16ac at I-PpoI-induced DNA breaks on human chromosome 1 and the *rDNA* locus (Fig. 5f and

Supplementary Fig. 6c) but not at the *GAPDH* locus (Supplementary Fig. 6d). Differences in Tip60 protein levels were not responsible for the defects observed in Tip60 recruitment when E2F1 was mutated or when E2F1 and RB were depleted (Fig. 5b and Supplementary Fig. 3a). Unlike p300 and CBP, no association was observed between Tip60 and phosphorylated E2F1 in the GST-TopBP1 pull-down assay (Supplementary Fig. 6e). This suggests that E2F1 may regulate Tip60 recruitment or retention at DNA breaks and induction of H4 acetylation through an indirect mechanism.

**The 3KR mutation prevents BRG1 and MRN accumulation at DSBs.** Our previous studies demonstrated that E2F1 and RB are required for the recruitment of a BRG1-containing nucleosome remodeling complex to DSBs and for decreasing nucleosome density at the sites of damage[7]. To determine whether E2F1 acetylation impacts this process, *E2f1³ᴷᴿ/³ᴷᴿ* knock-in MEFs and the I-PpoI ChIP assay system were employed. The E2F1 3KR mutation did not affect BRG1 protein levels (Supplementary Fig. 7a), but it did impair BRG1 recruitment to a DNA break (Fig. 6a). As we previously observed[7], E2F1-dependent recruitment of BRG1 is associated with decreased levels of total H3 protein at the site of damage, despite the increased levels of H3 acetylation marks (Fig. 6a).

It has been suggested that BRG1-containing complexes are recruited to DSBs through binding of the BRG1 bromodomain to acetylated histone H3 motifs[19]. Thus E2F1 acetylation may regulate BRG1 recruitment to DNA breaks by indirectly promoting p300/CBP-mediated H3 acetylation. On the other hand, we previously found that BRG1 physically associates with phosphorylated E2F1 and RB in response to DNA damage,

suggesting a direct role for E2F1 in recruiting BRG1 to sites of DNA breaks[7]. To differentiate between these mechanisms, the GST-TopBP1 pull-down assay was performed using *E2f1³ᴷᴿ/³ᴷᴿ* knock-in and wild-type control MEF extracts. The 3KR mutation did not prevent GST-TopBP1 from associating with E2F1 and RB in response to DNA damage, indicating that E2F1 phosphorylation and its interactions with TopBP1 and RB are unaffected by the 3KR knock-in mutation (Fig. 6b). As expected, the E2F1 3KR mutation prevented association of p300 with the GST-TopBP1-E2F1 complex (Fig. 6b). The ability of GST-TopBP1 to pull-down BRG1 was also compromised by the 3KR mutation, suggesting that E2F1 acetylation stabilizes the interaction between BRG1 and E2F1-RB (Fig. 6b). Indeed, BRG1 is known to physically interact with p300 in the context of transcription of E2F target genes[35].

The Mre11-RAD50-NBS1 (MRN) complex is a key DNA damage sensor and has roles in DNA end processing and in promoting the recruitment of various DNA repair proteins and chromatin-modifying activities to sites of damage. We previously observed that the absence of E2F1 impaired NBS1 foci formation in response to IR, suggesting that E2F1 is important for the recruitment and/or retention of MRN at DSBs[4]. Indeed, NBS1 and Mre11 were both enriched at an I-PpoI-induced DNA break in wild-type MEFs but not in *E2f1^{S29A/S29A}* or *E2f1³ᴷᴿ/³ᴷᴿ* MEFs (Fig. 6c). The E2F1 knock-in mutations did not affect the protein levels of Mre11 or NBS1 before or after DNA damage (Supplementary Fig. 7b). These findings confirm a role for E2F1 in the recruitment and/or retention of MRN at DNA breaks and indicates that both E2F1 phosphorylation and acetylation are important for this process.

NBS1 and Tip60 each participate in the activation of ATM at the sites of DNA damage[33,36–38]. Given that E2F1 and its

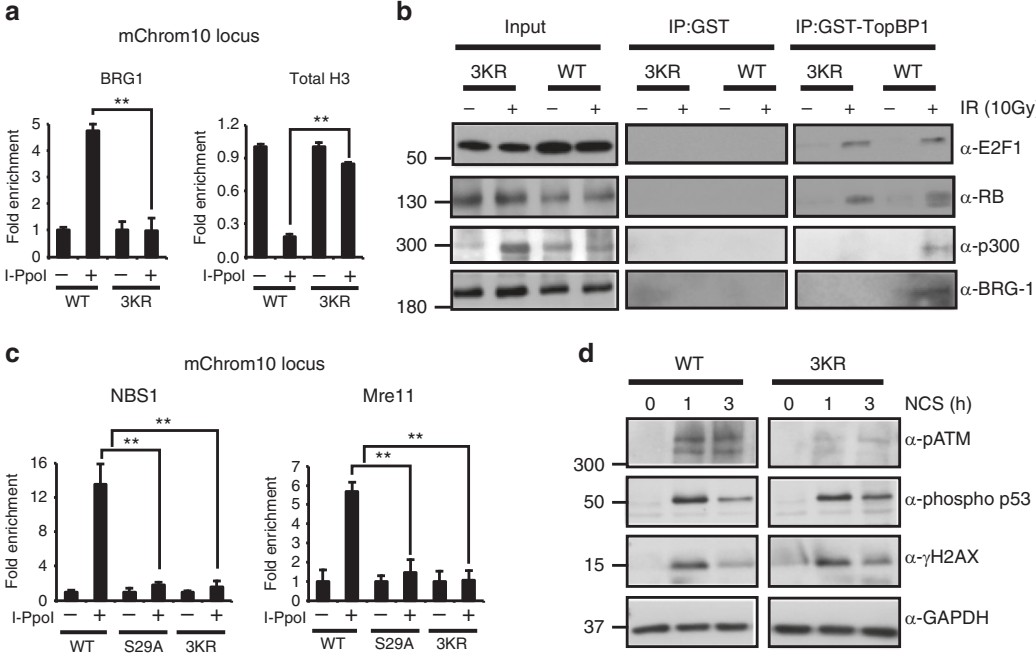

**Fig. 6** The E2F1 3KR mutation impairs BRG1 and MRN accumulation at a DSB. **a** Primary wild-type (WT) and *E2f1³ᴷᴿ/³ᴷᴿ* (3KR) MEFs were subjected to the I-PpoI ChIP assay and occupancy of BRG1 and total histone H3 was determined by qPCR at mChrom10 locus. **b** Purified GST-TopBP1 (BRCT1-6) or GST control protein was incubated with whole-cell extract from wild-type (WT) or *E2f1³ᴷᴿ/³ᴷᴿ* (3KR) MEFs that were either untreated (−) or treated (+) with IR (10 Gy) and harvested 2 h post-IR. Associated proteins were pulled down using glutathione beads and western blot was performed with the indicated antibodies. **c** Primary wild-type (WT), *E2f1^{S29A/S29A}* (S29A), and *E2f1³ᴷᴿ/³ᴷᴿ* (3KR) MEFs were subjected to the I-PpoI ChIP assay and occupancy of NBS1 and Mre11 was determined by qPCR at mChrom10 locus. **d** Western blot analysis was performed for phospho-ATM (serine 1981), phospho-p53 (serine 15), γH2AX, and GAPDH using whole-cell extracts from primary wild-type (WT) and *E2f1³ᴷᴿ/³ᴷᴿ* (3KR) MEFs, before treatment or 1 h and 3 h post-treatment with 250 ng/ml of NCS. All graphs represent average ± SD of three independent experiments (*n* = 3). *P* values were calculated by unpaired Student's *t* test. **\*\****P* ≤ 0.01 is highly significant. Source data of **b** and **d** are provided as Supplementary Data 5. Raw data of **a** and **c** are in Source Data File

posttranslational modifications are required for the accumulation of both factors at DSBs, it might be expected that ATM activation would be impaired by the E2F1 3KR knock-in mutation. As predicted, autophosphorylation of ATM at serine 1981, a marker of ATM activation[39], was significantly reduced in response to DNA damage in *E2f1*[3KR/3KR] mutant cells compared to wild-type cells although induction of p53 phosphorylation at serine 15 and H2AX phosphorylation at serine 139 (γH2AX) was not compromised by the 3KR knock-in mutation (Fig. 6d). This suggests that low levels of active ATM, or kinases related to ATM, are sufficient to induce normal levels of p53 and H2AX phosphorylation in *E2f1*[3KR/3KR] cells. Together, our results indicate that both E2F1 phosphorylation and acetylation are important for the recruitment of nucleosome remodeler BRG1 and MRN complex to DSBs.

**The E2F1 3KR knock-in mutation causes defective DSB repair**. As a read-out of DSB repair competency, we measured the kinetics of γH2AX and 53BP1 foci formation and clearance following IR exposure in wild-type and *E2f1*[3KR/3KR] MEFs. The 3KR mutation does not significantly affect γH2AX foci formation at earlier time points post-irradiation, but γH2AX foci clearance is significantly impaired at later time points in the knock-in compared to wild-type MEFs (Fig. 7a, b). Likewise, IR-induced 53BP1 foci formation in *E2f1*[3KR/3KR] MEFs is comparable to wild-type MEFs at early time points but 53BP1 foci clearance at later time points is significantly delayed (Fig. 7a, c). These findings suggest that the initial response to IR-induced DSBs is unaffected by the E2F1 3KR mutation but that later steps in the DSB repair process are impaired.

Cells defective for DSB repair are sensitized to chromosomal aberrations under genotoxic stress, which can lead to genome instability. To examine the impact of the E2F1 3KR mutation on chromosome maintenance, metaphase spreads were prepared from wild-type and *E2f1*[3KR/3KR] MEFs, 48 h post-IR. As expected, *E2f1*[3KR/3KR] MEFs had a significant increase in the percentage of cells with chromosome fusions, including dicentrics, rings, acentric, and long acrocentric chromosome-type aberrations (Fig. 7d). In addition, *E2f1*[3KR/3KR] metaphase spreads displayed increased numbers of small fragments and tetra/polyploid cells, which account for the significantly higher aberrant metaphases in the knock-in MEFs compared to wild-type MEFs (Fig. 7d). Of note, similar phenotypes of delayed DNA damage clearance and increased chromosome aberrations in response to genotoxic stress are also observed in *E2f1*[S29A/S29A] cells and in cells lacking E2F1 or RB[4,7].

We previously demonstrated that *E2f1*[S29A/S29A] mice are hypersensitive to IR, a hallmark of defective DSB repair[7]. To determine the functional significance of E2F1 acetylation in vivo, *E2f1*[3KR/3KR] and wild-type sibling control mice were irradiated with 5.5 Gy of IR and maintained under sterile conditions. The majority of *E2f1*[3KR/3KR] mice did not survive beyond 35 days following IR exposure, while 80% of wild-type siblings survived with no apparent signs of ill-health (Fig. 7e). Thus, like E2F1 phosphorylation, E2F1 acetylation may also play an important role in maintaining organismal homeostasis and survival in response to IR.

## Discussion

Prior studies have indicated a role for E2F1 acetylation in regulating its transcriptional activity, particularly the induction of *p73* gene expression in response to DNA damage[11,13,16]. However, in primary MEF cultures we find that *p73* gene expression is barely detectable and unresponsive to DNA damage. On the other hand, *p73* expression was induced in response to DNA damage in

several mouse tissues, although mutation of the three sites of acetylation in E2F1 either had no significant impact or actually enhanced this induction.

To gain a better understanding of how acetylation may regulate E2F1 transcriptional activity, we performed GSEA on RNA-seq data from primary MEFs derived from *E2f1*[3KR/3KR] and wild-type control mice, before and after the induction of DNA damage. Unexpectedly, >200 gene sets were significantly different between wild-type and *E2f1*[3KR/3KR] cells even in the absence of DNA damage. Gene sets that were enriched in wild-type cells compared to *E2f1*[3KR/3KR] cells, and thus likely to be positively regulated by E2F1 acetylation, were primarily related to nervous system development and differentiation (Supplementary Data 1 and 2). Importantly, gene sets related to DNA repair, DNA damage response signaling, cell proliferation, cell cycle checkpoints, or apoptosis were not significantly different between wild-type and *E2f1*[3KR/3KR] knock-in primary MEFs. This suggests that impaired DNA repair in *E2f1*[3KR/3KR] mutant MEFs, as indicated by delayed γH2AX and 53BP1 foci clearance and increased chromosomal aberrations following IR exposure, is not due to an indirect effect on transcription but rather on a direct effect of E2F1 at DSB sites. Interestingly, gene sets enriched in *E2f1*[3KR/3KR] MEFs were primarily involved in innate immunity and response to pathogens (Supplementary Data 3 and 4). Whether this is due to a direct transcriptional effect or an indirect consequence of genome instability caused by the *E2f1* knock-in mutation[40] is at present unclear.

E2F1 is recruited to DNA damage through a phospho-specific interaction with one of the BRCT domains of TopBP1[8]. We previously demonstrated that the RB tumor-suppressor protein associates with this phosphorylated form of E2F1 and is required for the stable interaction between E2F1 and TopBP1[7]. E2F1 and RB in turn recruit a BRG1-containing SWI/SNF complex to DSBs and this is associated with a decrease in total nucleosome density at the sites of damage[7]. We now show that an interaction between acetylated E2F1 and the bromodomains of p300 and CBP allows E2F1 and RB to directly recruit p300/CBP to sites of DNA breaks. Our data indicate that p300 and CBP then mediate the acetylation of multiple lysine residues on histone H3, including H3K18 and H3K56, in nucleosomes flanking DSBs. E2F1 and RB are also required for the recruitment of Tip60 and induction of H4K16ac at DNA breaks, although this may involve an indirect mechanism rather than a direct interaction between Tip60 and E2F1 or RB (Fig. 8).

Roles for p300, CBP, and Tip60 in the repair of DSBs are well established[19–21,23,27,32,33,41]. The p300 and CBP proteins were previously shown to localize to DSBs and to be required for H3K18 acetylation and the recruitment of SWI/SNF to sites of damage[20,21]. Knocking down p300 and CBP was also shown to impair H4 acetylation, the recruitment of DNA repair proteins to sites of damage, and DNA repair efficiency by both HR and non-homologous end-joining (NHEJ) pathways[20,21,25]. Findings presented here are consistent with those previous studies and now establish the molecular mechanism by which p300 and CBP are recruited to sites of DSBs dependent on RB and E2F1 post-translational modifications. This role for E2F1 in recruiting histone acetyltransferases to regulate DNA repair is reminiscent of its function in activating transcription, although the mechanisms by which E2F1 localizes to sites of DNA damage and target gene promoters are different[42–45].

Studies in the 1970s and 1980s demonstrated that histone acetylation and nucleosome remodeling occurs during the process of nucleotide excision repair (NER)[46,47]. Indeed, we previously demonstrated a role for E2F1 in the induction of histone acetylation and chromatin decondensation in response to ultraviolet radiation to promote efficient NER[3,48]. More recent studies are

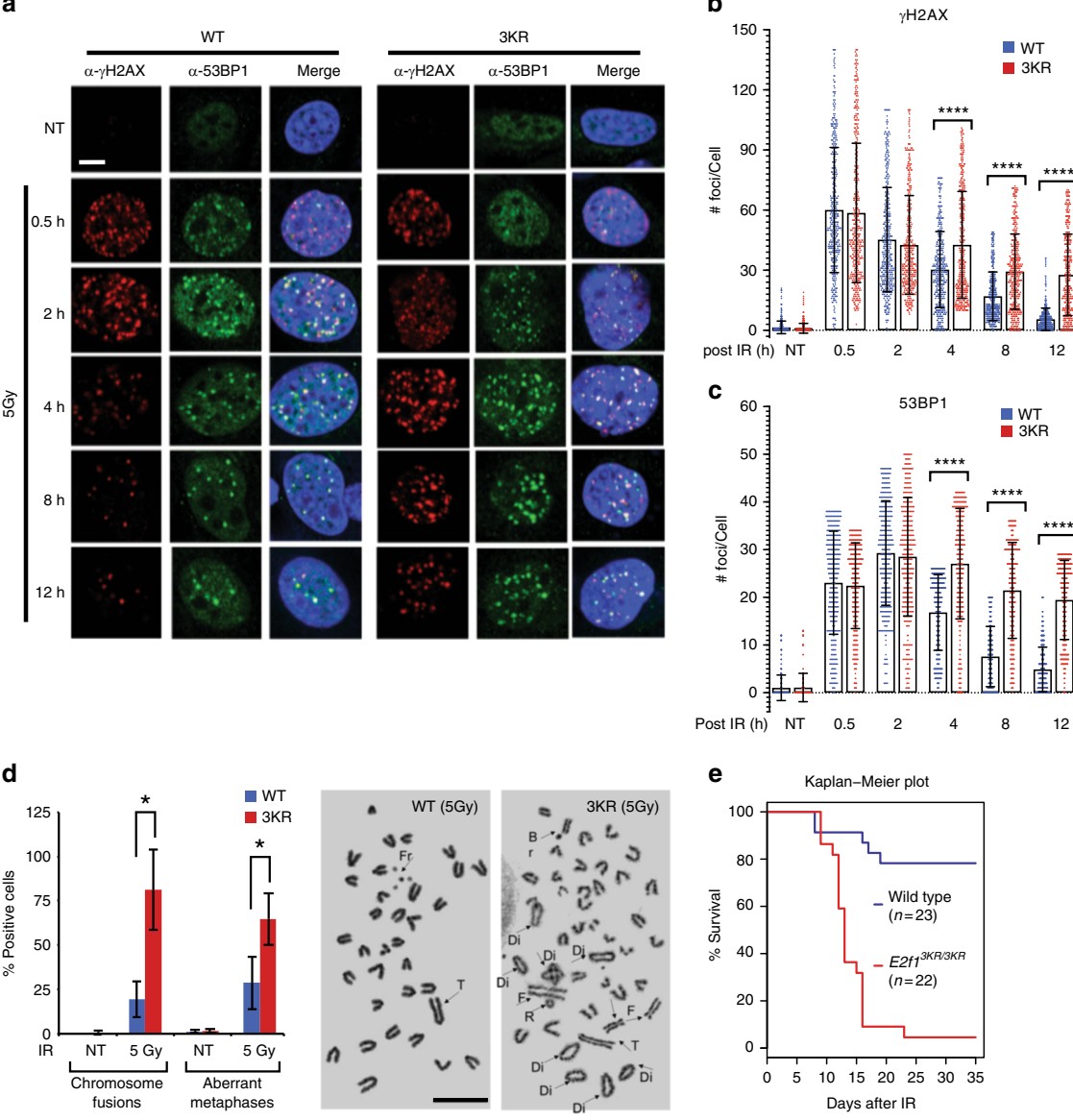

**Fig. 7** The 3KR mutation causes defective DNA repair phenotypes. **a** Primary wild-type (WT) and *E2f1^3KR/3KR* (3KR) MEFs were mock-treated (NT) or treated with 5 Gy of IR and allowed to recover for the indicated times. Representative confocal microscopic images show the formation and disappearance of γH2AX (red) and 53BP1 (green) foci and the merge with DAPI staining (blue) of nuclei. Bar, 10 μm. **b**, **c** Quantification of γH2AX foci (**b**) and 53BP1 foci (**c**) at the indicated times after IR was performed. Each treatment group was in triplicate, and in total at least 450 cells (n > 450) were counted per treatment group. Graphs represent average ± SD of foci per cell. Overlaid scattered dot plot shows the distribution of foci. *P* values were calculated by unpaired Mann–Whitney *U* test. ****P < 0.0001 is highly significant. **d** Primary wild-type (WT) and *E2f1^3KR/3KR* (3KR) MEFs were mock-treated (NT) or treated with 5 Gy of IR, and after 48 h, metaphase spreads were prepared. Chromosome fusions including dicentrics and rings and aberrant metaphases were counted and expressed as the percentage of positive cells. Each treatment group was in triplicate (n = 3), and at least 50 metaphases were counted per sample in the triplicate. Graphs represent average ± SD. *P* values were calculated by unpaired Student's *t* test. *P ≤ 0.05 is significant. Representative metaphase spreads are shown with arrows pointing to Fr—Fragment, Di—Dicentric, R—Ring, F—Fusion, Br—Break, and T—Tandem translocation. Bar, 10 μm. **e** Wild-type (n = 23) and *E2f1^3KR/3KR* (n = 22) mice were exposed to 5.5 Gy of IR and maintained for up to 35 days. Statistical significance in survival rates between genotypes was determined using the Kaplan–Meier method and *P* value was calculated by log-rank test. *P* < 0.0001 is highly significant. Raw data of averages in graphs underlying **b**–**e** are given in Source Data File

highlighting the role of dynamic chromatin remodeling in facilitating the efficient repair of DSBs[49,50]. These studies show that, immediately after a DNA break is induced, the surrounding chromatin undergoes a compaction involving the rapid recruitment of proteins, such as HP1, involved in mediating a repressive chromatin state[51–54]. However, this initial compact state is converted to an open, relaxed chromatin state within minutes and this requires ATM-dependent phosphorylation of downstream

targets, such as KAP1 and RNF20-RNF40[52,55–57]. This transition to an open chromatin state is also associated with Tip60-mediated H4 acetylation[33,50] and histone eviction at the DSB site[58].

We propose that p300/CBP-mediated H3 acetylation in nucleosomes surrounding a DSB also plays an important role in this transition from a compact to a relaxed form of chromatin, perhaps upstream or co-dependent with Tip60-mediated H4 acetylation. Histone acetylation likely cooperates with nucleolin

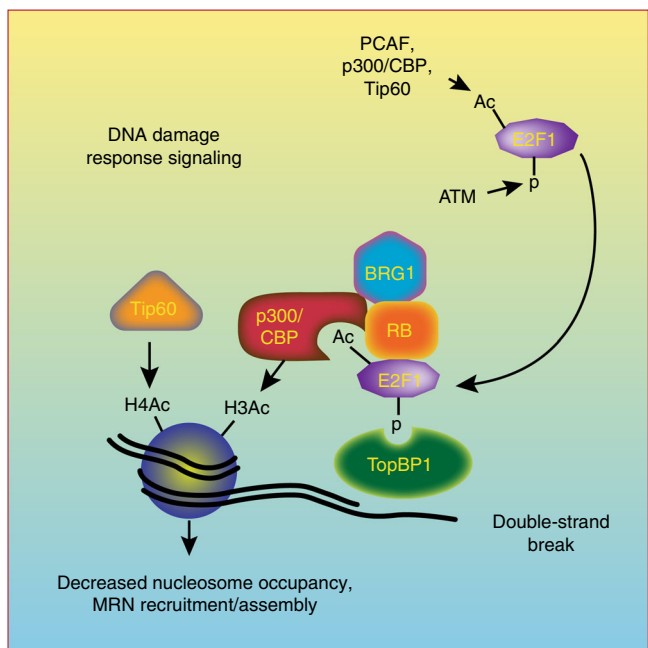

**Fig. 8** Posttranslational modifications regulate E2F1 recruitment and function at DSBs. E2F1 phosphorylation recruits E2F1 and RB to sites of DNA damage via an interaction with TopBP1, while E2F1 acetylation creates a binding motif for the bromodomains of p300 and CBP allowing E2F1 and RB to direct p300/CBP-mediated H3 acetylation in chromatin flanking DSBs. E2F1 phosphorylation and acetylation are also important for the recruitment of Tip60 and BRG1 to sites of DNA damage and for loading of the MRN complex

and nucleosome-remodeling complexes, such as SWI/SNF, to reduce nucleosome density at the sites of DNA damage to allow efficient access to the DNA repair machinery[19–21,58]. Thus E2F1 and RB may directly participate in relaxing chromatin structure at the sites of DNA damage by recruiting both BRG1-containing SWI/SNF complexes and p300/CBP.

RB is known to have multiple functions that help to preserve genome integrity independent of its role in regulating gene transcription, including maintaining heterochromatin structure at repetitive sequences and mediating chromosome condensation and cohesion during mitosis[59–62]. We and others have also demonstrated direct roles for RB and E2F1 in DNA break repair by both HR and NHEJ pathways[4,5,7]. Interestingly, it was recently revealed that the RB and E2F1 homologs of Arabipodsis, Retinoblastoma-Related (RBR) and E2FA, both co-localize with γH2AX foci in response to DNA damage in an ATM- and ATR-dependent manner[63,64]. As in humans and mice, RBR and E2FA participate in the recruitment of some DNA repair factors to the sites of damage to promote DNA repair and maintain genome integrity[63,64]. Whether Arabidopsis RBR and E2FA recruit histone acetyltransferases and/or nucleosome remodeling complexes to DSBs is currently unknown. Another open question is whether this direct role for E2F1 and RB homologs in DNA repair is conserved in other organisms.

We previously observed that NBS1 foci formation induced in response to DNA damage was significantly impaired when E2F1 was knocked out in mouse cells or knocked down in human cells[4]. This finding was confirmed and extended here using the I-PpoI ChIP assay system, which showed impaired accumulation of NBS1 and Mre11 at DSBs in both $E2f1^{S29A/S29A}$ and $E2f1^{3KR/3KR}$ knock-in MEFs. This indicates that both E2F1 phosphorylation and acetylation are important for the efficient loading of the MRN complex on chromatin flanking DNA break sites. The

MRN complex has multiple functions in DNA repair and DNA damage response signaling and its impaired accumulation at DSBs could explain several phenotypes observed in E2F1 mutant cells. The MRN complex is important for DNA end processing and resection to generate the single-stranded DNA required for HR repair as well as alternative NHEJ, also known as micro-homology repair[65]. NBS1 also directly binds ATM and partici-pates in the activation of ATM to amplify DNA damage response signaling[36,38], consistent with the reduced ATM autopho-sphorylation observed in E2F1 mutant cells. Moreover, the MRN complex participates in the eviction of histones at the sites of DSBs by directly binding to and recruiting the histone chaperone nucleolin[58].

It remains unclear exactly how E2F1 knock-in mutations impair the accumulation of MRN at DSB sites. Previous studies have demonstrated impaired accumulation of MRN at the sites of DNA damage in cells depleted for Tip60, and this was associated with defective histone acetylation[41,66]. This suggests that E2F1-dependent changes in chromatin structure may indirectly pro-mote the loading of MRN on chromatin flanking DSBs. On the other hand, E2F1 interacts with the N-terminus of NBS1, which could indicate a direct role for E2F1 in the recruitment and/or retention of MRN at the sites of DNA damage[67]. These possibi-lities are not mutually exclusive and E2F1 may promote MRN enrichment at DSBs by both direct and indirect mechanisms. Indeed, RB is also known to physically associate with several DNA repair proteins, including components of the NHEJ repair pathway[5]. These interactions, together with the E2F1–p300/CBP interaction defined here, could allow E2F1 and RB to coordinate the modification of chromatin structure with the assembly of DNA repair machinery at the sites of DSBs.

## Methods

**Generation of a targeted $E2f1^{3KR}$ knock-in allele.** A custom-made zinc finger nuclease (ZFN, Sigma) was designed to target mouse E2f1 sequences in exon 3 (AAACGGCGCATCTTATGACATCaccaatgtc, where the uppercase letters indi-cate the ZFN recognition sequence and lowercase letters represent the Folk1 cut site), approximately 70 bases 3' to the nearest codon encoding the three sites of E2F1 acetylation. A 1996 base pair donor construct was then designed (Genescript) to mutate codons 112, 115, and 120 (corresponding to human codons K117, K120, and K125 in humans) to alter protein coding from lysine to arginine (3KR mutation). One of these changes also resulted in the generation of unique Bgl II recognition site, as shown in Supplementary Fig. 4a. In addition, silent point mutations were also introduced into the donor construct to destroy the ZFN recognition sequence without affecting protein sequence. The isolated donor sequence fragment and mRNAs encoding the E2f1-targeting ZFN were given to the Transgenic Animal Facility at the University of Texas MD Anderson Cancer Center, Science Park for generating founder knock-in mice (FVB strain) by pro-nuclear injection. Genomic DNA from the resultant pups was isolated and PCR was used to clone E2f1 sequences that included the targeted codons and ZFN recognition site (forward primer: AGACATCAGACTGGGGTTGG, reverse pri-mer: TGACAGCAAAAGCTGGAATG). Targeted alleles were identified by diges-tion with Bgl II and sequencing of E2f1 exons 2–4 was performed to confirm the presence of the targeted mutations and absence of unintended mutations. A cor-rectly targeted founder mouse was crossed to a wild-type mouse and F1 mice positive for the $E2f1^{3KR}$ allele were further backcrossed to wild-type mice before heterozygous mice were crossed to generate homozygous $E2f1^{3KR/3KR}$ mice.

**Animal models.** Wild-type, $E2f1^{S29A/S29A}$, and $E2f1^{3KR/3KR}$ knock-in mice are FVB strains. The wild-type FVB mice (3–4 weeks old) are purchased (Harlan Labora-tories, IN) and the knock-in lines are produced in house. Generation of $E2f1^{S29A/S29A}$ knock-in mouse model was described earlier[3] and the strain details are deposited to Mouse Genome Informatics (MGI ID: 5755411) with a symbol of FVB.Cg-E2f1tm1.1Dgi. Development of $E2f1^{3KR/3KR}$ knock-in mouse model is described above and submitted to MGI (Symbol: E2f1em1Dgi, ID: 6313577).

**Maintenance of mice.** Wild-type, $E2f1^{S29A/S29A}$, and $E2f1^{3KR/3KR}$ knock-in mice of both sex from each strain up to age 1 year were maintained in an AAALAC-accredited facility in individually ventilated cages (Micro-isolator Housing Unit; Allentown Inc, NJ) sterilized together with aspen chip bedding (Aspen Sani-chips; P.J. Murphy Forest Products Corporation, NJ). Purina Irradiated Breeder Diet (Lab Diet #5058) was provided ad libitum and acidified reverse osmosis water was

provided by an automated system. Cages were maintained in a room where temperature and humidity were 20–22 °C and 55%, respectively, with a 14-h light and 10-h dark cycle and a minimum of 12 air changes per hour. Mice of both sex were randomly allocated to experiments at the age of 6–8 weeks. All animal experiments complied with the National Research Council's guide for the Care and Use of Laboratory Animals and were approved by the University of Texas MD Anderson Institutional Animal Care and Use Committee.

**Cell lines and primary cell cultures.** Primary MEFs of wild-type, $E2f1^{S29A/S29A}$, and $E2f1^{3KR/3KR}$ knock-in mice were isolated from 13.5-day-old embryos derived from crossing heterozygous or homozygous mice of each strain following standard procedures and maintained in Dulbecco's modified Eagle's medium (DMEM) supplemented with 10% heat-inactivated fetal bovine serum (FBS; Gibco), penicillin– streptomycin, and 100 μm β-mercaptoethanol at 37 °C in 5% $CO_2$ and 5% $O_2$. U2OS cells were obtained from American Type Culture Collection (ATCC, #HTB-96) and infected with lentiviral particles expressing short hairpin RNA (shRNA) targeting RB1 or E2F1[7]. U2OS parental and shRNA-expressing cell lines were maintained in DMEM supplemented with 10% FBS (Sigma) and penicillin–streptomycin at 37 °C in 5% $CO_2$.

**Treatment conditions and reagents.** IR treatment of human and mouse cell lines and of mice was performed with an RS-2000 biological irradiator (Rad Source) with the indicated doses. Where indicated, MEFs were treated with 1 μM p300/CBP bromodomain-specific inhibitor I-CBP112 and 250 ng/ml of radiomimetic drug NCS. The information of reagents and resources used in this study can be found in Supplementary Table 1 or in each method's section.

**Cador 5.0 protein domain microarray.** CADOR 5.0 chip[68] has 174 GST fusion proteins, arrayed in duplicate on a nitrocellulose slide. The layout of the array is shown in Supplementary Fig. 1a. The middle position (M) contains GST alone as a background indicator. The list of arrayed GST fusion proteins, accession numbers, and regions cloned can be found elsewhere[68,69]. The complementary DNAs (cDNAs) encoding the domains listed were cloned into the pGEX-6P1 vectors by PCR using a human cDNA library (Origene, MD) and verified by DNA sequencing[68]. Peptides were synthesized by W. M. Keck Biotechnology Resource Center (New Haven, CT) and CPC Scientific (Sunnyvale, CA). E2F1 peptides corresponding to human E2F1 amino acids (aa) 111–131 mono-acetylated at K117, K120, or K125; di-acetylated at 117/120, 117/125, and 120/125; tri-acetylated at each site (3KAc); or unacetylated (UnAc) were synthesized:

    E2F1 UnAc: biotin-RGRHPGKGVKSPGEKSRYETS-NH2
    E2F1 3KAc: biotin-RGRHPGK*GVK*SPGEK*SRYETS-NH2
    E2F1 K117Ac: biotin-RGRHPGK*GVKSPGEKSRYETS-NH2
    E2F1 K120Ac: biotin-RGRHPGKGVK*SPGEKSRYETS-NH2
    E2F1 K125Ac: biotin-RGRHPGKGVKSPGEK*SRYETS-NH2
    E2F1 K117/120Ac: biotin-RGRHPGK*GVK*SPGEKSRYETS-NH2
    E2F1 K117/125Ac: biotin-RGRHPGK*GVKSPGEK*SRYETS-NH2
    E2F1 K120/125Ac: biotin-RGRHPGKGVK*SPGEK*SRYETS-NH2

K* = acetylated lysine

Biotinylated peptides (10 μg) were pre-bound to 5 μl of Cy3–streptavidin or Cy5–streptavidin (Fluorolink™; Amersham Pharmacia Biotech) in 500 μl of phosphate-buffered saline (PBS)–Tween 20 (PBST). The fluorescently labeled peptide was then incubated with 20 μl of biotin–agarose beads to remove the free streptavidin label. Arrayed slides were blocked in PBST containing 3% (w/v) powdered milk, followed by the addition of 400 μl of fluorophore-tagged peptide. Blocking and hybridization were performed in an Atlas Glass Hybridization Chamber (Clontech). After incubation and washes, the slides were centrifuged to dry. Fluorescent signal was detected by GenePix 4200A Microarray Scanner (Molecular Devices), with the GenePix Pro Microarray Analysis Software (Molecular Devices). A 532 nm long pass filter was used for the detection of Cy3-labeled probes and fluorescein isothiocyanate-conjugated secondary antibodies. A 635 nm band pass filter was used for the detection of Cy5-labeled probes. A positive signal is seen as two dots at varying angles.

**Peptide pull-down assay.** Biotinylated peptides (15 μg) were immobilized on 25 μl of streptavidin beads in 500 μl of 1× mild buffer (50 mM Tris–HCl pH 7.5, 150 mM NaCl, 5 mM EDTA, 5 mM EGTA, 15 mM MgCl₂, and 0.1% NP-40), then incubated with 2–3 μg of GST fusion protein for 1 h. After washing, the beads were boiled in Laemmli buffer, separated by 10% sodium dodecyl sulfate (SDS)–polyacrylamide gel electrophoresis (PAGE) and subjected to western blot analysis using an anti-GST antibody.

**Purification of GST fusion proteins.** For NMR titration assays, GST fusion p300-BD (1040–1161 aa) and His-6x fusion p300-BRP (1051–1278 aa) proteins were expressed in minimal media supplemented with 50–150 μM ZnCl₂ and ¹⁵NH₄Cl. After induction with 0.4–1.0 mM isopropyl β-D-1-thiogalactopyranoside (IPTG) for 18 h at 16 °C, bacteria were harvested through centrifugation and lysed by sonication. GST fusion proteins were purified on glutathione-Agarose 4B beads (Thermo Fisher Sci) and the GST tag was cleaved with PreScission protease. His-tagged proteins were purified on Ni-NTA agarose beads and washed and eluted

with imidazole. The His-tag was removed by cleavage with PreScission protease. Proteins were concentrated using Millipore concentrators (Millipore).

For GST pull-down and peptide pull-down assays, indicated GST-fusion proteins were overexpressed in *Escherichia coli* DH5α cells (Life Technologies) by induction with a final concentration of 0.1 mM IPTG at 30 °C for 4 h. Cells were pelleted and resuspended in PBS, then subjected to sonication. The resulting lysates were centrifuged at maximum speed for 10 min, and the GST fusion proteins were then batch-purified from extracts by binding to glutathione–Sepharose 4B beads and washed in PBS according to the manufacturer's instructions. The purified proteins were eluted from the beads with elution buffer (30 mM Glutathione reduced, 100 mM Tris-HCl, pH 8.0, and 120 mM NaCl). The purified proteins were stored in the elution buffer at −80 °C.

**NMR experiments.** NMR experiments were performed at 298 K on a Varian INOVA 600 MHz spectrometer. ¹H,¹⁵N HSQC spectra of uniformly ¹⁵N-labeled p300 BD or BRP (0.1–0.15 mM) in buffer (BD: 25 mM Tris at pH 6.8, 150 mM NaCl, 2 mM dithiothreitol or BRP: 10 mM Hepes, 150 mM NaCl, 2% glycerol) and ~8% D₂O were collected as E2F1 peptides were added stepwise into the protein samples. NMR data were processed and analyzed with NMRPipe and NMRDraw[70]. NMRPipe is from NIST IBBR and available at https://www.ibbr.umd.edu/nmrpipe/index.html

**GST pull-down and IP blotting.** For GST pull-down experiments, cells were treated as indicated and harvested in cold PBS followed by resuspension in cell lysis buffer (20 mM Tris at pH 7.5, 150 mM NaCl, 1% Triton X-100, 1 mM EDTA, 1 mM EGTA, 2.5 mM Sodium pyrophosphate, protease inhibitor cocktail, phosphatase inhibitor, and HDAC inhibitors - 300 nM TSA, 2 μM EX-527). After aliquoting 5% of the lysates as input, 1–2 mg of lysates were pre-cleared by incubating with 25 μg of purified GST for 2 h at 4 °C and pulled down using glutathione-Sepharose 4B beads as per the manufacturer's directions. Pre-cleared lysates were divided equally to incubate with 15 μg of purified GST or GST-TopBP1 overnight at 4 °C. Next day, pull-down was performed using glutathione beads. Both samples and input were mixed with Laemmli buffer, boiled for 5 min, separated by SDS-PAGE, and subjected to western blot analysis[7]. See Supplementary Table 2 for the list of antibodies used.

For co-immunoprecipitation, cell lysates were prepared as mentioned above. Four hundred μg of lysates were pre-cleared, followed by IP with IgG control or 2.5 μg of antibody to E2F1 (C-20) and pulled down by protein G magnetic beads (Cell Signaling Technology). Immunocomplexes were eluted by boiling the beads with Laemmli buffer, followed by western blotting with acetylated lysine or E2F1. For western blot analysis of cell lysates, 15–100 μg of samples were mixed with Laemmli buffer, boiled for 5 min, run on SDS-PAGE, and then transferred onto a polyvinylidene difluoride membrane (Amersham Hybond). Blots were blocked in PBS–Tween 20 (PBST) or TBST containing 5% non-fat dry milk or bovine serum albumin respectively, and then incubated with primary antibody in the blocking buffer overnight at 4 °C. The list of antibodies used in this study is given in Supplementary Table 2. After washing and incubating with horseradish peroxidase (HRP)-conjugated secondary antibodies (Santa Cruz Biotechnology and Abcam), the membrane was subjected to ECL (Enhanced Chemiluminescence) detection as per the supplier's instructions. Signal was exposed to Autorad film (GeneMate) and developed using Medical Film processor (Konica Minolta SRX-101A).

**ChIP and qPCR.** Transduction of MEFs and U2OS cells with retrovirus expressing HA-ER*-I-PpoI enzyme was performed twice to increase infection efficiency and treated with 2 μM 4-hydroxy tamoxifen (Sigma) for 12 h to induce DNA damage[22]. Cells were crosslinked by adding formaldehyde (1% final concentration) followed by quenching with glycine (Sigma) at 1.25 mM final concentration, then harvested. Cell pellets were resuspended in SDS lysis buffer (1% SDS, 10 mM EDTA, 50 mM Tris at pH 8.1, protease inhibitor cocktail, phosphatase inhibitor, and HDAC inhibitors - 300 nM TSA, 2 μM EX-527) and the lysates were sonicated. After diluting with ChIP dilution buffer (0.01% SDS, 1.2 mM EDTA, 16.7 mM Tris at pH 8.1, 1.1% Triton X-100, 167 mM NaCl, protease inhibitor cocktail, phosphatase inhibitor, and HDAC inhibitors - 300 nM TSA, 2 μM EX-527), the lysates were pre-cleared and subjected to IP with the antibodies indicated (Supplementary Table 2). In all, 10% input was aliquoted separately. Next day, the DNA-bound immunocomplexes were pulled down by ChIP-Grade Protein G Magnetic beads and subjected to serial washes with buffers of low salt, high salt, LiCl, and TE. Then the crosslinked DNA was eluted with Elution buffer (1% SDS and 0.1 M NaHCO₃) at 65 °C. Subsequently, the samples including input were incubated with 5 M NaCl at 65 °C overnight for reverse crosslinking. After RNase A (Sigma) and Proteinase K (Sigma) treatment, the DNA was eluted using the QIAquick PCR Purification Kit (Qiagen) as per the manufacturer's instructions.

Occupancy of the DNA-bound proteins was measured by subjecting eluted DNA to qPCR using 7500 Fast Real-Time PCR system (Applied Biosystems) with primers for the indicated loci. Primers are listed in Supplementary Table 3. The percentage of input was calculated by dividing the amount of DNA obtained from the IP of the given factor by the total amount of DNA (input) and normalized for background signal (non-specific IgG control). Each experiment was carried out in triplicate and the results were expressed as relative enrichment, which represents

the ratio between the normalized percentage of input of infected cells and uninfected cells.

**IR treatment of mice and survival study post-IR.** Pups of wild-type and *E2f1³ᴷᴿ/³ᴷᴿ* strain were weaned at 21 days according to sex, then aged to 6–8 weeks prior to IR treatment. Mice were shifted to IR treatment room and transferred into sterile treatment cages in the biosafety cabinet. After 5.5 Gy of irradiation (up to 5 mice per treatment cage), they were returned to biosafety hood and transferred into sterile cages (with sterile feed, bedding, and water containing 50 mg/ml of Clava-mox). In the biosafety hood, antibiotic water bottles were changed out every 5 days and cages were changed out every 7 days for animals remaining housed after treatment. Mice were group housed if possible; single housed if need to prevent fighting. Mice were monitored daily for symptoms of ill-health and were eutha-nized if they became moribund as per federal and institutional guidelines.

**Cleaved Lamin A staining and immunohistochemistry (IHC).** Wild-type and *E2f1³ᴷᴿ/³ᴷᴿ* mice were subjected to IR treatment as described above and sacrificed 2 h later. Thymus and spleen tissues were collected, mounted on to cassettes, fixed in 10% neutral buffered formalin for 24–48 h, and then moved to 70% ethanol and paraffin embedded. Tissue sections were deparaffinized in xylene or xylene sub-stitute followed by graded alcohols (100%, 95% Ethanol) to water. Endogenous peroxidase activity was blocked with 3% $H_2O_2$ for 10 min in water. Antigen was retrieved with 10 mM Citrate Buffer at pH 6.0 in a microwave oven for 3 min at 100% power followed by 15 min at 50% power, then cooled down for 20 min. Binding of non-specific antibody was blocked by incubating the slides with blocking reagent (Biocare Medical) for 10 min. Then the slides were incubated with primary cleaved Lamin A antibody overnight at 4 °C (refer Supplementary Table 2 for antibody information). Next, these were washed twice and incubated with Envision plus labeled polymer and anti-rabbit-HRP (Dako) for 30 min at room temperature. After washing twice, these were incubated with DAB (Dako) for monitoring staining development. Finally, these were counterstained and dehy-drated, and coverslips were placed on the slides for viewing.

A set of 48 glass slides stained for IHC Lamin A were scanned by using the Aperio ScanScope imaging platform (Leica Biosystems, IL) with a ×20 objective at a spatial sampling period of 0.47 μm/pixel. After saving each digital image, a Genie classifier algorithm was trained to quantitate cleaved Lamin A-positive and -negative nuclei. Whole-slide images were viewed and analyzed by using desktop personal computers equipped with the free ScanScope software.

**RNA isolation and qPCR of gene expression.** Total RNA was isolated from wild-type and knock-in MEFs before and 3 h after treatment with NCS using the GenElute mammalian Total RNA Miniprep Kit (Sigma) following the supplier's instructions. For tissue samples, wild-type and *E2f1³ᴷᴿ/³ᴷᴿ* mice were subjected to 5.5 Gy of IR treatment and sacrificed 2 h later. Thymus, spleen, and liver tissues were collected and subjected to total RNA isolation using the same kit as per the instructions. On-column DNase I digestion was performed for all samples before eluting the RNA. Five hundred ng of total RNA was used to prepare cDNA using SuperScript II Reverse transcriptase with random hexamer primer. cDNAs were subjected to real-time qPCR with gene-specific primers. Primers are listed in Supplementary Table 3. Relative mRNA quantification was performed by com-parative $C_T$ method using *Gapdh* as an internal control.

**RNA-seq and GSEA.** Total RNA was isolated from wild-type and *E2f1³ᴷᴿ/³ᴷᴿ* MEFs before and 3 h after treatment with NCS using the GenElute mammalian Total RNA Miniprep Kit (Sigma) following the supplier's instructions. DNase I-treated RNA samples were fragmented and tagged at both ends for stranded library preparation using the TrueSeq Stranded mRNA Kit (Illumina). The mRNA-seq run was performed on the HiSeq 3000 platform (Illumina). Approximately 40–60 million reads were acquired per sample. The differential expression analysis for mRNA-seq data was performed with DESeq2 bioconductor R package with the cutoff of False Discovery Rate (FDR) $q \leq 0.05$. The normalized read count was generated from built-in function in DESeq2.

Sequenced tags were analyzed by GSEA downloaded from Broad Institute at http://software.broadinstitute.org/gsea/index.jsp. The mouse gene sets for Gene Ontology (GO) was prebuilt and can be downloaded from http://www.bioinformatics.org/go2msig/. The high-quality GO annotations for biological process GO terms with MsigDB format (.gmt) were utilized. Additional two input files, expression dataset file (.txt) and phenotype labels file (.cls), were generated according to the file formats described in GSEA user guide and GSEA data format guide.

**IF staining and confocal microscopy.** Wild-type and *E2f1³ᴷᴿ/³ᴷᴿ* MEFs ($0.5 \times 10^6$ cells) were seeded on 35 mm FluoroDish with 0.17 mm cover glass bottom (World Precision Instruments), 24 h prior to experiment. For γH2AX and 53BP1 foci formation and clearance assay, cells were treated with IR and, at the indicated times, rinsed with cold PBS, and fixed for 15 min at room temperature in 4% paraformaldehyde and 1% sucrose. For CBP and p300 IF experiments, cells were irradiated, harvested 2 h post-IR, and subjected to in situ extraction protocol as follows. After rinsing with cold PBS, cells were incubated 5 min on ice with pre-

extraction buffer (25 mM HEPES pH 7.5, 50 mM NaCl, 1 mM EDTA, 3 mM MgCl₂, 300 mM Sucrose, 0.5% Triton X-100), followed by 5-min incubation on ice with stripping buffer (10 mM Tris pH 7.4,10 mM NaCl, 3 mM MgCl₂, 1 mM EDTA, 1% Tween-20, 0.5% Na-Deoxycholate), and fixed as above.

After washing, cells were permeabilized with 0.5% Triton X-100 and blocked in Background Sniper for 15 min. Antibodies against CBP, p300, γH2AX, and 53BP1 were diluted with DaVinci Green antibody diluent and incubated with the cells overnight at 4 °C (See Supplementary Table 2 for antibody information). After washing, cells were incubated with appropriate fluorophore-conjugated secondary antibodies (Thermo Fisher Sci), stained with 4,6-diamidino-2-phenylindole and mounted in SlowFade Diamond Anti-fade mountant (Thermo Fisher Sci). Laser scanning confocal microscopy was performed using a Zeiss LSM880 and ×63 oil (1.4 NA) Plan/Apo objective with a pinhole aperture of 1–1.5 AU. Foci quantification of $5 \times 5$ tile scans were performed using the Imaris (Bitplane, v9.0) image analysis software. Cells module and custom algorithms specific to CBP, p300, γH2AX and 53BP1 foci. Each treatment group was in triplicate, and in total at least 450 cells were counted per treatment group. Images were captured and processed using identical microscope settings and foci parameters.

**Metaphase chromosome spread preparation.** Wild-type and *E2f1³ᴷᴿ/³ᴷᴿ* MEFs were irradiated as indicated and, 38 h after treatment, incubated with 0.1 μg/ml colcemid for 10 h. Cells were trypsinized, pelleted down, kept in a hypotonic solution (0.075 M KCl) for 15 min at room temperature, then fixed with 1:3 acetic acid:methanol, and spread on a slide. Metaphase chromosome aberrations of breaks, dicentrics, rings, chromosome fusions, fragments, and other aberrations were analyzed by Nikon Microscope with ×63 oil immersion objective. Images were taken using Imaging system from Applied Spectral Imaging (Carlsbad, CA). Approximately, 150 metaphases were counted per treatment. Each treatment group was set up in triplicate, and at least 50 metaphases were counted per sample in the triplicate.

**Statistical information.** For statistical analysis of mice survival curve, a Kaplan–Meier estimate was generated and analyzed for statistical significance with log-rank test. The difference in the survival rates between genotypes is highly significant with values of $P < 0.0001$. All other quantitative experiments were car-ried out in triplicate, and graphs represent average ± standard deviation (SD). Statistical analysis was performed using unpaired Student's $t$ test for all experi-ments except foci quantification by IF staining in which unpaired Mann–Whitney $U$ test was used. Significance was determined, with $P \leq 0.05$ (*) considered to be significant and $P \leq 0.01$ (**) and $P < 0.0001$ (****) are considered to be highly significant. For GSEA of RNA-seq data, significantly enriched pathways between genotypes were determined with cutoff of FDR value, $q \leq 0.05$.

**Reporting summary.** Further information on research design is available in the Nature Research Reporting Summary linked to this article.

## Data availability

Knock-in mouse models used in this study are registered in Mouse Genome Informatics (MGI) database. Strain details of *E2f1^{S29A/S29A}* (FVB.Cg-E2f1tm1.1Dgi) can be found in MGI ID:5755411 [http://www.informatics.jax.org/allele/MGI:5637520] and *E2f1³ᴷᴿ/³ᴷᴿ* (E2f1em1Dgi) in MGI ID:6313577 [http://www.informatics.jax.org/allele/key/882523]. RNA-seq data are available in Gene Expression Omnibus (GEO) repository under accession GSE135360. All data supporting the findings of this study are available within the article and its supplementary information files. Additional information and relevant data will be available from the corresponding author upon reasonable request. Supplementary Data 5 contains raw data of blots/gels underlying Fig. 1b, c; 2a, b; 4b, c; 5b; 6b, d; and Supplementary Figs. 1c; 2a; 3d; 4b; 6e; 7a, b. Source Data file contains raw data of all reported averages in graphs and charts underlying main figures, Fig. 3a–c; Fig. 4a, d and e; Fig. 5a, d–f; Fig. 6a and c; Fig. 7b–e; and Supplementary Figs. 2b, c; 3b, c and e; 4c; 5a–c and 6a–d.

## Code availability

Data in this manuscript are generated using commonly available commercial software and algorithms and are detailed in the corresponding "Methods" section. Specific computer code is not applicable.

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

## Acknowledgements

We thank B. Brooks and R. Deen for manuscript preparation; J. Holcomb for graphics; J. Orona and M. Portis for expert technical assistance; Ray. S for preliminary study; Lin. K for biostatistics; C. Sagum and the Protein Array and Analysis Core (supported by CPRIT RP180804) and J. Terpstra and C. Jeter of the Flow Cytometry & Cellular Imaging Core (supported by CPRIT RP170628) for assistance with IF; JJ Shen and the Next-Generation Sequencing Core (supported by CPRIT RP120348 and RP170002) for RNA-Seq; A. Multani and the Molecular Cytogenetics Facility for metaphase spreads analysis; D. Hollowell and the Transgenic Animal Core for generating knock-in mice; C. Perez of the Research Histology, Pathology and Imaging services for IHC; and Dale Weiss and colleagues in the Research Animal Support Facility for animal care. This work was supported by grants from the Cancer Prevention and Research Institute of Texas (RP140222 to D.G.J.); the National Institutes of Health (CA214723 to D.G.J., GM100907 to T.G.K. and Cancer Core Support Grant CA016672); and institutional funding from the Department of Epigenetics and Molecular Carcinogenesis, the Center for Cancer Epigenetics, and the Center for Genetics and Genomics.

## Author contributions

Conceptualization: S.M., R.V.-C., and D.G.J.; methodology: S.M., R.V.-C., A.K.B., M.T.B., and T.G.K.; investigation: S.M., R.V.-C., E.B., and B.J.K; formal analysis: B.L.; writing—original draft: S.M. and D.G.J., writing—review and editing: R.V.-C., A.K.B., M.T.B., and T.G.K.; funding acquisition: M.T.B., T.G.K., and D.G.J., supervision: M.T.B., T.G.K., and D.G.J.

## Competing interests

The authors declare no competing interests.
