## [Peer Review File · Nature Communications]

Reviewers' comments:

Reviewer #1 (Remarks to the Author):

Manickavinayaham et al "E2F1 Acetylation Directs p300/CBP-mediated Histone Acetylation at DNA Double-Strand Breaks to Facilitate Repair"

The manuscript identifies a detailed molecular pathway for E2F1 acetylation in the response to DNA double strand breaks. The authors show that E2F1 acetylation, which is induced by DNA damage, is recognized by the HATs p300 and CBP via the highly related bromodomains of these two enzymes. This leads to more stable association of p300 and CBP with chromatin, histone acetylation (H3K18 and H3K56), which in turn leads to the recruitment of additional factors involved in resolving DSBs. The study is comprehensive, ranging from NMR analysis of bromodomain binding to acetylated-E2F1 peptides and protein array experiments to metaphase spreads and mouse models with Kaplan-Meier survival curves in response to irradiation. Overall this is exciting and important work that should be of high interest to the broad readership of Nature Communication.

Comments:

- 1) The authors write that the bromodomains of CBP and p300 specifically bind to acetylated E2F1. It would be helpful to elaborate on or clarify what is meant by specificity. Do CBP and p300 bind with selectivity to acetylated E2F1 sites versus other acetylated sites or is it that other bromodomains do not bind to acetylated E2F1?
- 2) The specificity of the H3K56Ac antibody has been questioned. Do the authors have evidence for the specificity of their anti-acetyl antibodies (anti-H3K56Ac and anti-H3K18Ac)? H3K56Ac per se is likely not central to the overall model, so if the antibody specificity cannot be resolved, a brief discussion of this issue would be helpful.

Reviewer #2 (Remarks to the Author):

In this manuscript Manickavinayaham et al investigate the functional significance of three acetylation sites (K117, K120, K125) within the amino-terminal domain of E2F1. They report that these residues are acetylated by p300 and CBP, that the interaction between E2F1 and p300 and CBP recruits these enzymes to double strand breaks and promotes Histone acetylation at the breaks. They show that mutation of these sites impairs an interaction between TOPBP1 and BRG1 and reduces the recruitment/retention of NBS1 and MRN to DNA breaks, impeding repair. An important feature of this study is that Manickavinayaham et al. have generated a knock-in allele in which three lysine residues of E2F1 have been mutated to arginine. Manickavinayaham et al. report the transcriptional changes that they detect in these animals and, in addition to the molecular changes at break sites, they show that MEFs from these animals have defects in the repair of double strand breaks and IR-induced chromosomal aberrations. They show that mice homozygous for the knock-in allele are hypersensitive to irradiation.

This is a very well written manuscript on an important topic in E2F research. It has become clear that E2F and RB proteins have activities that are different from the classical view of these proteins as transcriptional regulators. When compared with the vast literature on transcriptional properties of E2F proteins, the number of studies of these non-canonical functions is small. More information is needed about the molecular mechanisms and their functional significance. One of the clearest examples of the non-canonical E2F/RB functions was work published from the Johnson group describing the recruitment of E2F1 and RB to sites of double strand breaks. This manuscript

represents a very important and interesting extension of this work.

This study draws together several previous observations and provides much needed clarity about the significance and impact of E2F1 acetylation. What makes this paper special is the generation of the mutant E2F1 allele. Others have previously described acetylation of E2F1 and its interaction with p300 and CBP and have proposed roles for this modification in transcriptional regulation. By generating the knock-in allele Manickavinayaham et al provide key information about the significance of these interactions, and the molecular events at sites of DSB that occur as a result of E2F1/RB recruitment. The data is convincing and the models that the authors build are persuasive.

While the data that Manickavinayaham et al present is very consistent with their interpretation that the functional defects that they describe in the knock-in mice are due to the direct action of E2F1 at sites of double strand breaks I suspect that there will be some in the E2F community who will cling to the idea that the mutant phenotypes are, at least in part, the result of transcriptional changes even though the authors state that there are no good candidates for this connection. I must say though that the data in the paper that that surprised me the most is Figure 7e. The change in animal survival is striking and this kind of animal phenotype is not usually shown in DDR papers that typically use cell-based assays. I really don't know how to interpret this data and would appreciate any information that the authors can provide to support their interpretation that this lethality is due to the direct action of E2F1 in DSB repair. Can they provide example of other mutant mice in DDR pathways that show a similarly strong in vivo phenotype? Perhaps my worries are unfounded, but I am concerned that this phenotype is too strong (the result is too good), and that the lethality may reflect changes in gene expression programs and not just the proposed direct role at double strand breaks.

Reviewer #3 (Remarks to the Author):

In the manuscript "E2F1 Acetylation Directs p300/CBP-mediated Histone Acetylation at DNA Double- Strand Breaks to Facilitate Repair" by Manickavinayaham and collaborators, the authors set out to address the role of E2F1 acetylation in the DNA damage response and DNA damage repair.

By means of a biochemical screen, they identify CBP and p300 as two acetyltransferases capable of binding to an E2F1 peptide only when this is acetylated on three lysines: 117, 120 and 125, with a prominent contribution played by lysines 125 and 117. By further biochemical characterization they show that these three lysines are required for the recruitment of both CBP and p300 on DSBs, which in turn, by acetylating chromatin on H3K18 and H3K56, favour the recruitment of Tip60, BRG1, NBS1 and MRE11 therefore allowing efficient repair of damaged sites. Accordingly, homozygous E2F-3KR knock-in cells and mice display defective DNA repair, chromosomal instability and radiosensitivity.

Overall the work is novel and provides sufficient evidence to support the model proposed. Major shortcomings are the lack of in vivo evidence supporting that indeed E2F1 is acetylated on those residues following DNA damage and stronger experimental evidences showing that E2F1-3K is impaired in the recruitment of CBP/p300 on DNA damaged sites.

Following are few points which the Authors will need to address in order to strengthen their manuscript:

Major Points

1. Can the Author provide evidence for DSB dependent acetylation of E2F1? If not feasible, I would ask the Authors to provide a detailed explanation of how they tried to assess this.
2. While biochemical studies point to a major role of two lysines (K117 and K125), knock-in studies were only performed with the E2F1-3KR mutant, where the three lysines (177, 120 and 125) are mutated. Can the Authors clarify if there is any contribution of lysine 120? Will a E2F1-117/125KR mutant be comparable to the E2F1-3KR? What is the effect of mutations in the single residues?
3. Beyond ChIP experiments, Authors should show the colocalization of E2F1 and CBP/p300 (or lack of it, in the case of E2F1-3KR) by alternative methods: coIP, colocalization by immunofluorescence, or proximity ligation assay.
4. The model presented predicts that the function of "acetylated"-E2F1 is to allow recruitment of CBP/p300, which in turn will trigger the recruitment of Tip60, BRG1, NBS1 and MRE11 to favour repair. If this is correct, inhibition of CBP/p300 by CBP112 should phenocopy E2F1-3KR (i.e. should prevent the recruitment of Tip60, BRG1, NBS1, MRE11, decrease DNA repair and make cells more radiosensitive). This should be tested experimentally.
5. Concerning the E2F1-3KR knock-in, the effect on DNA damage induced apoptosis is mild (fig4d), yet radio-sensitivity is greatly enhanced (fig 7e), Why? I suggest to evaluate apoptosis also in Bone Marrow and in the Intestinal Mucosa, where modulation of DNA-damage induced apoptosis usually accounts for radiosensitive phenotypes and differences in the apoptotic index might be more in line with the strong radiosensitization observed in E2F1-3KR knock-in animals.

Minor point:

- (1) Please, indicate a reference for the acetylated residues and briefly explain the rationale for testing only peptides containing those 3 lysines.

Reviewers' comments:

Reviewer #1 (Remarks to the Author):

Manickavinayaham et al "E2F1 Acetylation Directs p300/CBP-mediated Histone Acetylation at DNA Double-Strand Breaks to Facilitate Repair"

The manuscript identifies a detailed molecular pathway for E2F1 acetylation in the response to DNA double strand breaks. The authors show that E2F1 acetylation, which is induced by DNA damage, is recognized by the HATs p300 and CBP via the highly related bromodomains of these two enzymes. This leads to more stable association of p300 and CBP with chromatin, histone acetylation (H3K18 and H3K56), which in turn leads to the recruitment of additional factors involved in resolving DSBs. The study is comprehensive, ranging from NMR analysis of bromodomain binding to acetylated-E2F1 peptides and protein array experiments to metaphase spreads and mouse models with Kaplan-Meier survival curves in response to irradiation. Overall this is exciting and important work that should be of high interest to the broad readership of Nature Communication.

Comments:

1) The authors write that the bromodomains of CBP and p300 specifically bind to acetylated E2F1. It would be helpful to elaborate on or clarify what is meant by specificity. Do CBP and p300 bind with selectivity to acetylated E2F1 sites versus other acetylated sites or is it that other bromodomains do not bind to acetylated E2F1?

What is meant by specificity in this context is p300 and CBP bromodomains binding to acetylated E2F1 with higher affinity compared to unacetylated E2F1. This specificity is evident from the protein domain microarray, the GST pull down assays and the NMR chemical shift analysis. Results from the protein domain microarray screen demonstrate that, compared to the unacetylated E2F1 peptide, the acetylated E2F1 peptide binds with greater affinity to the p300 bromodomain, but not to bromodomains from other proteins, including GCN5, PCAF, BRDT, BAZ, TAF1, SNF2, and BAF180. Follow up experiments demonstrated that, like p300, the CBP bromodomain also bound to the acetylated E2F1 peptide with higher affinity compared to the unacetylated peptide, but this was not true of the TAF1 or BRD9 bromodomains. (Figure 2b of manuscript and data not shown). Results from the heteronuclear single quantum coherence NMR titration experiments suggest that the acetylated E2F1 peptide binds to the isolated p300 bromodomain with a K_d of at least 185 +/- 74 μ M and to a recombinant p300 construct containing the bromodomain + PHD domain with a K_d of at least 88 +/- 17 μ M. This is similar to the binding affinities of other known acetylated motifs read by the CBP and p300 bromodomains, such as p53K382Ac and H3K36Ac^{1,2}. We did not include these K_d values in the manuscript, because the peptides used in the chemical shift analysis were considered of low purity by our collaborator and, thus, the numbers above are an underestimate of the actual values.

2) The specificity of the H3K56Ac antibody has been questioned. Do the authors have evidence for the specificity of their anti-acetyl antibodies (anti-H3K56Ac and

anti-H3K18Ac)? H3K56Ac per se is likely not central to the overall model, so if the antibody specificity cannot be resolved, a brief discussion of this issue would be helpful.

We are aware of the reported issues with some commercially available H3K56Ac antibodies. According to a study from the Tyler laboratory³, the H3K56Ac antibody from Cell Signaling Technology (Cat# 4243) that we used in this study is specific for H3K56Ac in a peptide dot blot array experiment, unlike other H3K56Ac antibodies from companies such as Epitomics, Epigentec, and Upstate. It was demonstrated that a H3K27Ac peptide can compete for this H3K56Ac antibody from Cell Signaling Technology, so it is possible that it cross-reacts with the H3K27Ac mark. As the reviewer points out however, this issue is not central to the main conclusions of the paper as p300 and CBP are also known to acetylate H3K27.

Reviewer #2 (Remarks to the Author):

In this manuscript Manickavinayaham et al investigate the functional significance of three acetylation sites (K117, K120, K125) within the amino-terminal domain of E2F1. They report that these residues are acetylated by p300 and CBP, that the interaction between E2F1 and p300 and CBP recruits these enzymes to double strand breaks and promotes Histone acetylation at the breaks. They show that mutation of these sites impairs an interaction between TOPBP1 and BRG1 and reduces the recruitment/retention of NBS1 and MRN to DNA breaks, impeding repair. An important feature of this study is that Manickavinayaham et al. have generated a knock-in allele in which three lysine residues of E2F1 have been mutated to arginine. Manickavinayaham et al. report the transcriptional changes that they detect in these animals and, in addition to the molecular changes at break sites, they show that MEFs from these animals have defects in the repair of double strand breaks and IR-induced chromosomal aberrations. They show that mice homozygous for the knock-in allele are hypersensitive to irradiation.

This is a very well written manuscript on an important topic in E2F research. It has become clear that E2F and RB proteins have activities that are different from the classical view of these proteins as transcriptional regulators. When compared with the vast literature on transcriptional properties of E2F proteins, the number of studies of these non-canonical functions is small. More information is needed about the molecular mechanisms and their functional significance. One of the clearest examples of the non-canonical E2F/RB functions was work published from the Johnson group describing the recruitment of E2F1 and RB to sites of double strand breaks. This manuscript represents a very important and interesting extension of this work.

This study draws together several previous observations and provides much needed clarity about the significance and impact of E2F1 acetylation. What makes this paper special is the generation of the mutant E2F1 allele. Others have previously described acetylation of E2F1 and its interaction with p300 and CBP and

have proposed roles for this modification in transcriptional regulation. By generating the knock-in allele Manickavinayaham et al provide key information about the significance of these interactions, and the molecular events at sites of DSB that occur as a result of E2F1/RB recruitment. The data is convincing and the models that the authors build are persuasive.

While the data that Manickavinayaham et al present is very consistent with their interpretation that the functional defects that they describe in the knock-in mice are due to the direct action of E2F1 at sites of double strand breaks I suspect that there will be some in the E2F community who will cling to the idea that the mutant phenotypes are, at least in part, the result of transcriptional changes even though the authors state that there are no good candidates for this connection. I must say though that the data in the paper that that surprised me the most is Figure 7e. The change in animal survival is striking and this kind of animal phenotype is not usually shown in DDR papers that typically use cell-based assays. I really don't know how to interpret this data and would appreciate any information that the authors can provide to support their interpretation that this lethality is due to the direct action of E2F1 in DSB repair. Can they provide example of other mutant mice in DDR pathways that show a similarly strong in vivo phenotype? Perhaps my worries are unfounded, but I am concerned that this phenotype is too strong (the result is too good), and that the lethality may reflect changes in gene expression programs and not just the proposed direct role at double strand breaks.

First, it should be pointed out that in Figure 7e we use an IR dose of 5.5 Gy of total body irradiation, which seems to be the "sweet spot" for demonstrating a difference between wild type and E2F1 mutant mice. Previously, we observed that increasing the IR dose results in more deaths in the wild type group (FVB strain background), while decreasing the dose results in more survival of E2F1 mutant strains (knockout and S29A knock-in). Similar IR hypersensitivity phenotypes are observed in NBS1 hypomorphic mutant mice⁴ and 53BP1 mutant mice⁵, among others. Hypersensitivity of E2F1 mutant mice appears to be less severe than the phenotypes of *Atm* and *Ku80* knockout mice^{6,7}.

Reviewer #3 (Remarks to the Author):

In the manuscript "E2F1 Acetylation Directs p300/CBP-mediated Histone Acetylation at DNA Double- Strand Breaks to Facilitate Repair" by Manickavinayaham and collaborators, the authors set out to address the role of E2F1 acetylation in the DNA damage response and DNA damage repair.

By means of a biochemical screen, they identify CBP and p300 as two acetyltransferases capable of binding to an E2F1 peptide only when this is acetylated on three lysines: 117, 120 and 125, with a prominent contribution played by lysines 125 and 117. By further biochemical characterization they show that

these three lysines are required for the recruitment of both CBP and p300 on DSBs, which in turn, by acetylating chromatin on H3K18 and H3K56, favour the recruitment of Tip60, BRG1, NBS1 and MRE11 therefore allowing efficient repair of damaged sites. Accordingly, homozygous E2F-3KR knock-in cells and mice display defective DNA repair, chromosomal instability and radiosensitivity.

Overall the work is novel and provides sufficient evidence to support the model proposed. Major shortcomings are the lack of *in vivo* evidence supporting that indeed E2F1 is acetylated on those residues following DNA damage and stronger experimental evidences showing that E2F1-3K is impaired in the recruitment of CBP/p300 on DNA damaged sites.

Following are few points which the Authors will need to address in order to strengthen their manuscript:

Major Points

1. Can the Author provide evidence for DSB dependent acetylation of E2F1? If not feasible, I would ask the Authors to provide a detailed explanation of how they tried to assess this.

There is not a commercially available antibody specific for the acetylated form of E2F1 and we have been unsuccessful in developing such an antibody. A number of reports previously demonstrated that E2F1 is acetylated in response to DNA damage using other methods⁸⁻¹². Most groups monitored E2F1 acetylation by first immunoprecipitating total E2F1 followed by western blot analysis of the precipitated proteins using an antibody to acetylated lysine⁹⁻¹¹. We have now repeated this experiment and have obtained similar results indicating that E2F1 acetylation increases in response to DNA damage but this is blocked by the 3KR mutation. This new data is now included in the revised manuscript as Figure 4c.

2. While biochemical studies point to a major role of two lysines (K117 and K125), knock-in studies were only performed with the E2F1-3KR mutant, where the three lysines (177, 120 and 125) are mutated. Can the Authors clarify if there is any contribution of lysine 120? Will a E2F1-117/125KR mutant be comparable to the E2F1-3KR? What is the effect of mutations in the single residues?

We have not made knock-in mouse alleles with individual or double lysine acetylation sites mutated so we cannot compare the phenotypes of 3KR mice/cells to individual or double mutant cells/mice. We have extended our *in vitro* binding studies to examine binding of the p300 and CBP bromodomains to E2F1 peptides acetylated on two of the three lysine residues. In the case of the p300 bromodomain, acetylation at K125 was critical for binding to an E2F1 peptide, and additional acetylation at K117 and/or K120 had no apparent effect on binding. The CBP bromodomain appeared to bind equally well to an E2F1 peptide mono-acetylated at either K117 or K125, and combining those acetylation marks, or adding K120 acetylation, did not enhance binding between E2F1 and CBP. These new results suggest no contribution of K120 acetylation to CBP or p300

bromodomain binding and are now included in the supplemental results section of the revised manuscript.

3. Beyond ChIP experiments, Authors should show the colocalization of E2F1 and CBP/p300 (or lack of it, in the case of E2F1-3KR) by alternative methods: coIP, colocalization by immunofluorescence, or proximity ligation assay.

In addition to demonstrating by ChIP the colocalization of E2F1 and CBP/p300 at four different genomic loci (two in mouse and two in human cells) with induced DNA double-strand breaks, we have also demonstrated a physical association between E2F1 and CBP/p300 that is enhanced in response to DNA damage. The GST-TopBP1 pull down experiments shown in Figures 2A, 2B, and 6B are essentially co-IP experiments except that instead of an antibody, GST-TopBP1 is used as a reagent to precipitate phosphorylated E2F1 and its associated proteins. Results from these experiments demonstrate a physical association between endogenous E2F1 and CBP/p300 in wild type cells, but not in 3KR cells. Below are two older experiments in which Flag-tagged versions of E2F1 (either wild type or 3KR mutant) were transfected into human cells and western blot analysis was performed on Flag-immunoprecipitated proteins (see Figure 1 below). As expected, wild type Flag-E2F1 associated with both CBP and p300 in response DNA damage while the 3KR mutant failed to do so. While these experiments with overexpressed, exogenous Flag-tagged E2F1 further support our finding of a physical interaction between E2F1 and p300/CBP in response to DNA damage, we do not feel they add to the existing data with endogenous E2F1 and have therefore chosen not to include this data in the manuscript.

We have also performed new immunofluorescence (IF) experiments as proposed by the reviewer and find that in wild type cells, p300 and CBP form IR-induced foci that partially colocalize with γ H2AX, indicating their recruitment to double-strand breaks. In sharp contrast, p300 and CBP did not form IR-induced foci in MEFs harboring the E2F1 3KR mutation. This new IF data is included in the revised manuscript as Figures 5c, 5d, and Supplemental Figure 5c. We also renewed our efforts to detect E2F1 foci by IF in mouse cells to demonstrate co-localization with p300/CBP. We tested three new batches of E2F1 antibody (Santa Cruz (C-20) #sc-193, Abcam #ab179445, and Santa Cruz (KH95) #sc-251). These antibodies worked well in western blots but, as we previously observed, not for IF.

4. The model presented predicts that the function of “acetylated”-E2F1 is to allow recruitment of CBP/p300, which in turn will trigger the recruitment of Tip60, BRG1, NBS1 and MRE11 to favour repair. If this is correct, inhibition of CBP/p300 by CBP112 should phenocopy E2F1-3KR (i.e. should prevent the recruitment of Tip60, BRG1, NBS1, MRE11, decrease DNA repair and make cells more radiosensitive). This should be tested experimentally.

We agree with the reviewer that these are interesting experiments, and in the future, we plan to examine CBP112 as a potential radiosensitizer. However, we anticipate that inhibition of the CBP and p300 bromodomains will have a wider impact on CBP and p300 functions than just blocking E2F1-dependent recruitment to DSBs. For instance, the CBP bromodomain is known to bind to acetylated p53 (K382Ac) to regulate p53 transcriptional activity, p21 expression and cell cycle arrest¹. Thus, we have limited our studies with CBP112 to examining recruitment of p300/CBP and early chromatin changes (at 12h) at DSB sites. Further studies examining downstream effects of CBP112 on DNA repair and radiosensitivity would require additional RNA-seq experiments to rule out possible indirect effects caused by transcriptional changes in DNA repair and damage response genes. This was done for the E2F1 3KR mutation, which we feel is a cleaner system for examining the function of the E2F1-p300/CBP interaction in cells.

5. Concerning the E2F1-3KR knock-in, the effect on DNA damage induced apoptosis is mild (fig4d), yet radio-sensitivity is greatly enhanced (fig 7e), Why? I suggest to evaluate apoptosis also in Bone Marrow and in the Intestinal Mucosa, where modulation of DNA-damage induced apoptosis usually accounts for radiosensitive phenotypes and differences in the apoptotic index might be more in line with the strong radiosensitization observed in E2F1-3KR knock-in animals.

As suggested, we examined IR-induced apoptosis in the intestine and found that the 3KR mutation has no significant impact on apoptosis levels in this tissue 24 hr after IR exposure, a time when levels of intestinal apoptosis are at their peak (see Figure 2 below). It should be pointed out that mouse radiosensitivity does not correlate with the acute apoptotic response in tissues. A good example is *Atm* knockout mice, which are highly radiosensitive but display reduced levels of apoptosis in response to IR in at least some tissues^{6,13}.

Minor point:

1) Please, indicate a reference for the acetylated residues and briefly explain the rationale for testing only peptides containing those 3 lysines.

In 2000, two independent groups reported that E2F1 can be acetylated by PCAF, p300 and CBP and mapped the acetylated sites to K117, K120, and K125 of human E2F1^{12,14}. It was later reported that Tip60 could also acetylate E2F1 at K120 and 125, but not K117¹¹. None of these studies found evidence of E2F1 acetylation at other sites.

References:

1. Mujtaba, S. *et al.* Structural mechanism of the bromodomain of the coactivator CBP in p53 transcriptional activation. *Mol. Cell* **13**, 251-263, (2004).
2. Zeng, L., Zhang, Q., Gerona-Navarro, G., Moshkina, N. & Zhou, M. M. Structural basis of site-specific histone recognition by the bromodomains of human coactivators PCAF and CBP/p300. *Structure* **16**, 643-652, (2008).
3. Pal, S. *et al.* The Commercial Antibodies Widely Used to Measure H3 K56 Acetylation Are Non-Specific in Human and Drosophila Cells. *PLoS ONE* **11**, e0155409, (2016).
4. Kang, J., Bronson, R. T. & Xu, Y. Targeted disruption of NBS1 reveals its roles in mouse development and DNA repair. *EMBO J.* **21**, 1447-1455, (2002).
5. Morales, J. C. *et al.* Role for the BRCA1 C-terminal repeats (BRCT) protein 53BP1 in maintaining genomic stability. *J. Biol. Chem.* **278**, 14971-14977, (2003).
6. Barlow, C. *et al.* *Atm*-deficient mice: a paradigm of ataxia telangiectasia. *Cell* **86**, 159-171, (1996).
7. Nussenzweig, A., Sokol, K., Burgman, P., Li, L. & Li, G. C. Hypersensitivity of Ku80-deficient cell lines and mice to DNA damage: the effects of ionizing radiation on growth, survival, and development. *Proc. Natl. Acad. Sci. U. S. A.* **94**, 13588-13593, (1997).
8. Galbiati, L., Mendoza-Maldonado, R., Gutierrez, M. I. & Giacca, M. Regulation of E2F-1 after DNA damage by p300-mediated acetylation and ubiquitination. *Cell Cycle* **4**, 930-939, (2005).
9. Ianari, A., Gallo, R., Palma, M., Alesse, E. & Gulino, A. Specific role for PCAF acetyltransferase activity in E2F1 stabilization in response to DNA damage. *J. Biol. Chem.* **279**, 20830-20835, (2004).
10. Pediconi, N. *et al.* Differential regulation of E2F1 apoptotic target genes in response to DNA damage. *Nat. Cell Biol.* **5**, 552-558, (2003).
11. Van Den Broeck, A., Nissou, D., Brambilla, E., Eymin, B. & Gazzeri, S. Activation of a Tip60/E2F1/ERCC1 network in human lung adenocarcinoma cells exposed to cisplatin. *Carcinogenesis* **33**, 320-325, (2012).
12. Martinez-Balbas, M. A., Bauer, U. M., Nielsen, S. J., Brehm, A. & Kouzarides, T. Regulation of E2F1 activity by acetylation. *The EMBO Journal* **19**, 662-671, (2000).
13. Herzog, K. H., Chong, M. J., Kapsetaki, M., Morgan, J. I. & McKinnon, P. J. Requirement for *Atm* in ionizing radiation-induced cell death in the developing central nervous system. *Science* **280**, 1089-1091, (1998).
14. Marzio, G. *et al.* E2F family members are differentially regulated by reversible acetylation. *J. Biol. Chem.* **275**, 10887-10892, (2000).

Figure 1. E2F1 associates with CBP and p300 in response to DNA damage dependent on sites of E2F1 acetylation. 293T cells were transfected with plasmids expressing wild type Flag-E2F1 or Flag-E2F1 3KR mutant. Cells were untreated or exposed to 5Gy of IR 2 hr before harvesting. Anti-Flag antibody was used to immunoprecipitate exogenous E2F1 and western blot analysis was performed for a) CBP and b) p300.

Figure 2. IR-induced apoptosis in the intestine of E2F1 wild type and 3KR mutant mice. Wild type (WT) and *E2f1*^{3KR/3KR} (3KR) mice were exposed to 5.5 Gy of IR and intestinal tissues were collected 24h later. Tissue sections were stained for the cleaved form of lamin A and the average % of positive cells from three mice per group is presented +/- SD.

REVIEWERS' COMMENTS:

Reviewer #1 (Remarks to the Author):

The authors have addressed my comments and I recommend publication.

Reviewer #2 (Remarks to the Author):

The authors have addressed my comments.

This is a convincing study and, for the reasons outlined in the initial review, it is an important addition to the literature.

Reviewer #3 (Remarks to the Author):

The Authors provided satisfactory answers to most of my concerns. There are still a couple of minor amendments that should be implemented in order to make the manuscript fully suitable for publication:

Line 41: To my knowledge, reference 8 does not show that E2F1 co-localizes to DDR foci. If I am correct, this sentence should be amended or eliminated from the text.

Suppl figure 1: it is not clear what bait was used for the pull down of the GST fusions. A detailed description of the experiment should be included in the supplementary methods section.

The Answer to my "minor point" should be included in the text to help the reader in understanding the rationale for the selection of the acetylation sites analysed in this manuscript.

Point by Point Response to Reviewers Comments.

REVIEWERS' COMMENTS:

Reviewer #1 (Remarks to the Author):

The authors have addressed my comments and I recommend publication.

No response required.

Reviewer #2 (Remarks to the Author):

The authors have addressed my comments.

This is a convincing study and, for the reasons outlined in the initial review, it is an important addition to the literature.

No response required.

Reviewer #3 (Remarks to the Author):

The Authors provided satisfactory answers to most of my concerns. There are still a couple of minor amendments that should be implemented in order to make the manuscript fully suitable for publication:

Line 41: To my knowledge, reference 8 does not show that E2F1 co-localizes to DDR foci. If I am correct, this sentence should be amended or eliminated from the text.

Response: We thank the reviewer for catching this mistake. We have now replaced reference 8 with the correct reference, which was reference 9 in the previous version of the manuscript. The revised manuscript now has references 8 and 9 switched.

Suppl figure 1: it is not clear what bait was used for the pull down of the GST fusions. A detailed description of the experiment should be included in the supplementary methods section.

The bait in Supplementary Figure 1 is the same bait as in Figures 1b and 1c. Biotinylated peptides corresponding to human E2F1 amino acids 111-131, either acetylated or unacetylated on amino acids K117, K120, and/or K125 were used to pull down GST-p300 or GST-CBP as indicated above each lane. The E2F1 peptide sequences can be found in the materials and methods section. The bait used has now been clarified in the Supplementary Figure 1 legend in the revised manuscript.

The Answer to my “minor point” should be included in the text to help the reader in understanding the rationale for the selection of the acetylation sites analysed in this manuscript.

We describe the three sites of E2F1 acetylation and their regulation in the second paragraph of the introduction section and include primary references for this previous work. In response to the reviewer’s comments we now briefly summarize the rationale for examining these E2F1 acetylation sites in a new sentence at the beginning of the results section in the revised manuscript.